



# The Representation of Climate Impacts in the FRIDAv2.1 Integrated Assessment Model

Christopher Wells[1], Benjamin Blanz[2], Lennart Ramme[3], Jannes Breier[4], Beniamino Callegari[5], Adakudlu Muralidhar[6], Jefferson K. Rajah[7], Andreas Nicolaidis Lindqvist[8,9], Axel E. Eriksson[4,9,10], William Alexander Schoenberg[7,11], Alexandre C. Köberle[12], Lan Wang-Erlandsson[9,13,14], Cecilie Mauritzen[15], Chris Smith[16,17]

[1]School of Earth and Environment, University of Leeds, Leeds, LS2 9JT, United Kingdom
[2]Research Unit Sustainability and Climate Risks, University of Hamburg, Grindelberg 5, 20144 Hamburg, Germany
[3]Max-Planck-Institute for Meteorology, Bundesstraße 53, 20146 Hamburg, Germany
[4]Department of Earth System Analysis, Potsdam Institute for Climate Impact Research, Telegrafenberg A31, 14473 Potsdam, Germany
[5]School of Economics, Innovation and Technology, Kristiania University of Applied Sciences, Oslo, Norway
[6]Department of Ocean and Ice, Norwegian Meteorological Institute, 0313 Blindern, Oslo
[7]System Dynamics Group, University of Bergen, P.O. Box 7802, 5020 Bergen, Norway
[8]RISE Research Institutes of Sweden, Ideon Beta5, Scheelevägen 17, 22370, Lund, Sweden
[9]Stockholm Resilience Centre, Stockholm University, Albanovägen 28, SE-106 91 Stockholm
[10]Department of Environmental and Energy Systems Studies, Lund University, Box 118, 221 00 Lund
[11]isee systems inc., 24 Hanover St, Ste 8A, Lebanon, NH 03766 USA
[12]Instituto Dom Luiz, Faculty of Sciences, Universidade de Lisboa, Campo Grande, Edifício C1, Piso 1, 1749-016, Lisboa, Portugal
[13]Potsdam Institute for Climate Impact Research, Member of the Leibnitz Association, 14473 Potsdam, Germany
[14]Anthropocene Laboratory, the Royal Swedish Academy of Sciences, 104 05 Stockholm, Sweden
[15]Climate Department, Norwegian Meteorological Institute, 0313 Oslo, Norway
[16]Department of Water and Climate, Vrije Universiteit Brussel, 1050 Brussel, Belgium
[17]Energy, Climate and Environment Program, International Institute for Applied Systems Analysis (IIASA), Laxenburg, Austria

*Correspondence to*: Christopher Wells (c.d.wells@leeds.ac.uk)

**Abstract.** Feedbacks from the climate to other components of the coupled human-Earth system are expected to strongly influence the co-evolution of human society and its environment. Representing these feedback loops between climate and society, via the Earth system's response to human activities and the subsequent effect back onto social systems, is essential in order to fully explore the dynamics of the coupled system. However, focus on these feedbacks has traditionally been limited, or excluded, in prior Integrated Assessment Models (IAMs) and IAM-based modelling protocols. This limits the understanding of the effects of climate change, and the response of the overall system to future emissions scenarios and policies.

The new IAM Feedback-based knowledge Repository for IntegrateD Assessments version 2.1 (FRIDAv2.1), documented and explored within this collection, seeks to address this by internalising the feedbacks between subcomponents of the



human-Earth system. Within this new IAM, these connections are therefore a key part of the structure, and are documented and discussed here.

These feedbacks from the climate to human societies, conceptualised as climate impacts, are represented as global impact functions within FRIDA. Where possible, they are based on estimates from existing literature, reframed as functions of global climate variables to facilitate their representation within FRIDA. Other impact channels, with insufficient background literature to inform their structure and parameter values, are incorporated via the internal calibration of the IAM.

   Since the systematic representation of climate damages within an IAM is a relatively novel endeavour, the approach is
constrained by literature limitations and necessary simplifications. In addition, the high level of abstraction of the FRIDA model imposes limits on the set of impacts which can reasonably be implemented, and the level of process detail amongst those included. Nevertheless, FRIDA's endogenous representation of climate feedbacks to human and natural systems enables valuable insights and intuition building on an underexplored topic. The general nature of the climate damage functions aggregated and documented here allows for their incorporation within other models and frameworks.

**1 Introduction**

   The new fully coupled FRIDAv2.1 Integrated Assessment Model, presented in (Schoenberg et al., submitted.) and detailed further within this collection, seeks to place each component of the human-Earth system on an equal footing in terms of process complexity and interactions with other components. In this way, interactions between the Climate module of FRIDA and other modules represent climate impacts, broadly conceived. These climate impact interactions are expected to be key
drivers of the development of the full system. This paper details the climate impacts represented in FRIDAv2.1, documenting the connections between outputs from the Climate module and inputs to other modules of the model.

   FRIDA's model boundary encompasses the entire human-Earth system, and as such must sacrifice process complexity for the sake of legibility and tractability, as well as to ensure similar levels of detail in different domains, which is crucial for systems modelling (Robinson et al., 2018). One consequence of this is that version 2.1 of FRIDA remains a global model,
with no spatial disaggregation of its stocks and flows. "Climate impact channels" within FRIDA are therefore represented at this scale. Impact channels defined here reflect individual conceptual connections between a variable relating to climate and one which resides in a different domain.

   Many theoretical and observed linkages exist between the climate and social systems, with the Intergovernmental Panel on Climate Change (IPCC) Sixth Assessment Report (AR6) Working Group II (WGII) detailing a wide range of impacts in its
Technical Summary (Pörtner et al., 2022), and Ripple et al., (2023) collating 41 such processes. Complex connections have been studied across several domains and been found to be crucial drivers of system behaviour (e.g. Moore et al., 2022; Robinson et al., 2018), playing important roles in the modelling of socio-environmental systems (Elsawah et al., 2020).

   In implementing climate impacts in FRIDAv2.1, the set of impacts described in the IPCC WGII AR6 Technical Summary (Pörtner et al., 2022), representing the state-of-the-art comprehensive literature on climate impacts, were considered for



inclusion. As discussed in Section 3, while many impact channels are included in FRIDAv2.1, many more were not feasible
      for incorporation into this model, generally due to inapplicability to the scope of FRIDA as currently conceived.

      The representation of climate impacts, as expected, has a strong influence on climate policy measures such as the Social Cost
      of Carbon (SCC; Rennert et al., 2022); for example Li et al., (2020) found a change in the cost-optimised policy pathway
      upon inclusion of sea level rise (SLR) impacts. Global-scale modelling of climate impacts is made challenging by the

interactions between different spatial and temporal scales, as well as between modelling disciplines (Baldos et al., 2023).

      Estimates of the total impact of climate change often attempt to measure or model the overall impact on global Gross
      Domestic Product (GDP; see Howard & Sterner, (2017)). This is often framed in a top-down approach, whereby historical
      aggregate data is used to generate GDP damage functions, as opposed to a bottom-up process-based modelling approach, in
      which the specific processes driving impacts are studied and their total effect analysed.

Top-down empirical studies have found substantial projected climate damages (e.g. Burke et al., 2015; Kotz et al., 2022,
      2024; Waidelich et al., 2024). However, debates on whether climate impacts reduce GDP amounts (level effects) or their
      growth rates (growth effects) persist, with many studies modelling the lower damages associated with level effects with no
      persistence (e.g. Kotz et al., 2024; Waidelich et al., 2024), while Bastien-Olvera et al., (2022) have detected non-zero growth
      effects empirically.

Due to their somewhat piecemeal nature, bottom-up estimates can miss out impact channels (Howard and Sterner, 2017),
      leading generally to lower total impacts than top-down approaches (Shiogama et al., 2022). However, empirical analyses
      based on reported extreme events are also complicated by spatially-dependent data limitations (Newman and Noy, 2023),
      which can lead to underestimates of total climate impacts; and the modelled drivers of GDP impacts in top-down studies
      neglect several impact channels such as cyclones, as well as any effect of tipping points (Kotz et al., 2024). In bottom-up

approaches, care should be taken not to double-count impacts, particularly when global mean surface temperature change is
      used as a proxy for other climate variables.

      In the FRIDA integrated assessment model (IAM), the drive for transparency and understanding of specific feedback loops
      necessitates a process-based, multi-channel impact framework rather than a single GDP effect. In keeping with other
      representations in process-based IAMs (Diaz and Moore, 2017), this allows for the validation of the model's structure, a key

process in the system dynamics methodology (Barlas, 1996). A crucial step in the model creation process is therefore the
      selection of the specific impact channels modelled, since missing impact channels will lead to an underestimation of climate
      impacts. For FRIDAv2.1, the process of including climate impacts involved reviewing the literature on overall climate
      impacts to identify important channels (e.g. Pörtner et al., (2022)), with subjective importance assessed as related to a
      channel's prominence in the literature, followed by channel-specific literature analyses to determine how to account for each

impact.

      Several studies used older impact models, fed by climate model output, to explore the estimated sector-based impacts of
      climate change (e.g. Arnell et al., 2016a, b). This two-step approach – using climate model output to drive sector-specific
      impact models – has been systematically implemented and improved by successive generations of the Inter-Sectoral Impacts





Model Intercomparison Project (ISIMIP). Several ISIMIP phases, in conjunction with phases of climate scenario modelling,
have analysed the climate impacts of the Coupled Model Intercomparison Project Phase 5 (CMIP5) scenarios associated with
the Fifth IPCC assessment report (Frieler et al., 2017), and more recently those of CMIP6 (Frieler et al., 2024). Within each
modelling round, climate output from multiple Earth System Models, bias-corrected to more closely reflect climate
observations (e.g. Lange, 2021), is fed into multiple impact models for a given sector. This allows for a systematic multi-
model analysis of the sectoral impacts of climate change, and is designed to generate impact functions for use in IAMs such
as FRIDA.

The split between changes in climate (as modelled in Earth System Models) and the effect of these changes on specific
sectors (as simulated by impact models) is enshrined in the framework of the Shared Socioeconomic Pathways (SSPs),
which simulated potential future demographic and emissions pathways without feedbacks from climate impacts (O'Neill et
al., 2014). This framework separates climate changes from climate impacts to avoid double counting when feeding climate
model output into impact models, which have a more detailed representation of impact processes than the scenario-
generating IAMs (van Vuuren et al., 2025). While the limitations of this framework are widely acknowledged, and calls have
been made to integrate impacts more closely within the framework (Pirani et al., 2024), the next generation of scenarios
towards CMIP7 will likely feature the same split, with the emissions scenarios not including the effect of climate damages
(Meinshausen et al., 2024; van Vuuren et al., 2025).

The continuation of this split between climate change and impacts motivates the development and use of fully coupled IAMs
like FRIDA. Such models are instrumental for exploring the linkages between climate and society under future scenarios,
and understanding how societal actions, occurring concurrently with ongoing climate change, may significantly modify its
trajectory. It should also be noted that the level and distribution of climate impacts depends on the baseline socioeconomic
conditions (Pirani et al., 2024; Takakura et al., 2019), further motivating the exploration of this coupled system.

The framework used in ISIMIP has also been modified to incorporate rapid emulation of climate model output, allowing for
substantially faster simulation of climate impacts, though only for a single impact model (Mathison et al., 2025). Coupled
human-climate interactions more broadly have also been studied within agent-based modelling frameworks, a distinct
approach to the one in FRIDA (Lamperti et al., 2018; Rounsevell et al., 2012).

Several studies have suggested the existence of tipping points within the Earth system, with strongly nonlinear impacts upon
reaching a given temperature threshold, and potential hysteresis in the response to a peak-and-decline scenario (Diaz and
Moore, 2017; McKay et al., 2022), although the tipping point conceptual framework has been critiqued (Kopp et al., 2025).
The specific thresholds and responses are uncertain, and the spatial pattern of these impacts further complicates their
inclusion in a global model. FRIDAv2.1 does not include explicit tipping behaviour in its climate module, though the
interconnected nature of the components allows for strongly nonlinear behaviour; this is further discussed in Section 3.

Section 2 describes the specific implementation of the climate impacts represented in FRIDAv2.1, with additional detail in
the appendix where necessary. In general, for each identified impact channel, relevant literature was explored to identify





possible routes for inclusion into FRIDA. Where possible, uncertainties are given for the 95 percent confidence (2.5th-97.5th percentile) range. The methodology and the broader context are discussed in Section 3, and Section 4 provides conclusions.

## 2 Climate Impact Channels in FRIDAv2.1

This section describes the implementation of the climate impact channels in FRIDAv2.1. Figure 1 shows the set of climate impacts represented in FRIDA, indicating their drivers in the Climate module and the module impacted by each channel. All six non-climate modules in FRIDA – Energy, Behavioural Change, Resources, Demographics, Economy, and Land Use & Agriculture – are affected by one or more climate impact channel(s).

Table 1 provides a textual summary of impact channels considered in FRIDA, and the associated literature sources,
functional form, treatment of uncertainty, and parameter values. Each channel is presented and discussed in Section 2. The resultant functions are shown in Fig. 1 and Fig. A1, with the functional forms detailed in Section A.2.

The methodology for selecting climate impact channels for inclusion within FRIDAv2.1 involved the systematic consideration of the climate impacts detailed in the Technical Summary to the WGII report of the IPCC AR6 (Pörtner et al., 2022). These impacts were categorised into three broad areas: those necessary for prioritised inclusion within FRIDAv2.1;
those which could plausibly be represented in a model such as FRIDA, but which reflected a lower priority, whether due to the perceived magnitude of their effect or the timescale and detail required to facilitate their implementation; and finally those whose inclusion in a highly aggregated model such as FRIDA was not deemed feasible. The first category - the climate impacts represented in FRIDAv2.1 - is documented in this section; Section 3.1 explores the other two.





**Figure 1: Outputs from the Climate module (left), and their linkages to other modules in FRIDAv2.1 (right) via climate impact channels. Coloured lines and groupings indicate the module affected by each set of impacts; grey lines inside the central boxes show the specific drivers of each damage. Each numbered box represents a distinct climate impact channel, with its implementation process described in the corresponding part of Section 2. Graphs to the right of each channel name show the damage function(s) implemented as part of that impact channel; see Fig. A1 for a more detailed version of these graphs. Within the graphs, red curves represent the median central estimate, with grey giving the uncertainty variation. In cases in which uncertainty is treated by "Sampled Percentiles" in Table 1, the 11 parameter sets representing the 2.5th to 97.5th percentiles, with equal uncertainty spacing, are shown; for the other impact channels, the range using the upper and lower parameter values after literature analysis and internal calibration is shown. Completion of the human-Earth system feedback loops, via the cascading effect on human drivers of the Earth system response, is represented schematically by the grey arrow. \*Sea Level Rise impacts are indicated in this figure, but generally cannot be reduced to a single function, and therefore are not represented in the plots. Instead these are discussed in their respective sections and Ramme et al. (submitted).**

Where possible, data to generate impact functions were taken from existing literature, either from modelling or empirical observation studies. Several impact channels, included in FRIDA due to their assessed importance in order to aid internal consistency, had no identified appropriate literature. These were therefore incorporated with functional forms - which were expert supplied and ensured to be dimensionally consistent, with the extreme condition response validated - and the



parameter values calibrated internally, as described in this paper as it relates to climate impact channels, and with the internal calibration method described in (Schoenberg et al., submitted.). Under uncertainty analysis in FRIDAv2.1, Uniform samples are taken of the parameter ranges given in Table 1 for these channels.

For literature-based estimates, where possible, uncertainty was re-sampled to provide 11 parameter sets equally spaced from the 2.5th to 97.5th percentiles (designated as "Sampled Percentiles" in the Uncertainty column of Table 1). This was done to capture the impacts of covariance between parameter values within each impact channel, and to allow for the independent varying of uncertainties within future uncertainty analysis. Only the median values are given in Table 1; for information on the 11 parameter sets used, see the Code and Data Availability section.

Climate damages in FRIDA are driven by a combination of three climate outputs: global mean surface temperature (GMST), Sea Level Rise (SLR), and atmospheric $CO_2$ concentrations. The novel process-based SLR model in FRIDA, termed FRISIA, is documented in Ramme et al. (submitted). In FRISIA, the five most important contributors to SLR (thermal expansion, land water storage changes, and separate melting of mountain glaciers and the Greenland and Antarctic ice sheets) are modelled individually and calibrated to match observations and IPCC estimates including their uncertainty ranges

(Fox-Kemper et al., 2021; Horwath et al., 2021). The rising sea level then causes damage along the global coast through a set of different impact strains. However, the strength of those impacts is highly dependent on the socio-economic evolution of the coastal zone. For this reason, the model of SLR impacts in FRIDAv2.1 specifically tracks stocks of coastal assets and population. While Ramme et al. (submitted) discuss the quantification of the impacts of SLR and how this model relates to other studies of SLR impacts, here the impact channels themselves are described, in their respective sections.

In contrast to other climate impacts in FRIDA, the possibility of detailed adaptation against SLR damages is incorporated, both because there is historical evidence of adaptation measures against storm surge damages, and since it has been shown that adaptation can substantially reduce future SLR-driven damages (IPCC, 2022; Tiggeloven et al., 2020; Wong et al., 2022). However, in FRIDA's baseline run configuration, proactive adaptation against SLR is turned off, in line with the expected adaptation based on historical action against other climate impacts in FRIDA.





| Category | Variable | Functional form | Uncertainty treatment | Parameter values | Reference |
|---|---|---|---|---|---|
| Energy supply (Section 2.1.1) | Hydropower energy efficiency | Quadratic in GMST anomaly (Form 1) | Sampled Percentiles* | a = -0.5316 %K$^{-1}$  b = -0.2470 %K$^{-2}$ | Van Vliet et al., (2016) |
|  | Thermoelectric energy efficiency |  |  | a = -2.731 %K$^{-1}$  b = -0.5341 %K$^{-2}$ |  |
| Energy demand (Section 2.1.2) | Cooling Degree Days (CDD) | Linear in GMST anomaly (Form 2 with intercept) | Internally calibrated in FRIDA | a = 704.2 K day  b = 287.85 (287.01 - 288.69) day | Werning, Frank, et al., (2024), Werning, Hooke, et al., (2024) |
|  | Heating Degree Days (HDD) | Exponential in GMST anomaly |  | a = 1376 K day  b = -0.2326 (-0.2405 to -0.2246)K$^{-1}$ |  |
|  | Cooling energy change due to climate change | Linear in CDD | Internally calibrated to match literature | a = 0.000875 (0.0004 - 0.01) GJ (CDD)$^{-1}$ | Rode et al., (2021), Clarke et al., (2018) |
|  | Cooling energy change due to climate change | Linear in HDD |  | a = 0.002 (0.0001 - 0.01) GJ (HDD)$^{-1}$ |  |
| Energy infrastructure | Damage rate of capital | Power law in GMST anomaly | Internally calibrated in | a = 0.0000501 (0 - 0.0001019) yr$^{-1}$ | N/A |



| damage (Section 2.1.3) | | | FRIDA | b = 2 (1 or 3) | |
|---|---|---|---|---|---|
| Exposure & Climate Risk Perception (Section 2.2.1) | Historical record-breaking extremes exposure per person per year | Quadratic in GMST anomaly (Form 1) | Sampled Percentiles* | a = 0.7207 Indices person$^{-1}$yr$^{-1}$K$^{-1}$<br><br>b = -0.02483 Indices person$^{-1}$yr$^{-1}$K$^{-2}$ | (Li et al., 2023) |
| Durability of Concrete (Section 2.3.1) | Average lifetime of concrete | Linear in GMST anomaly (Form 2) | Approximated from literature | -7.5 (-5 to -15) %K$^{-1}$ | Bastidas-Arteaga & Stewart, (2015), Stewart et al., (2011), Wang et al., (2012) |
| Mortality (Section 2.4.1) | Cold impact on mortality | Quadratic in GMST anomaly (Form 1) | Sampled Percentiles* | a = -1.8993 %K$^{-1}$<br><br>b = 0.1941 %K$^{-2}$ | Bressler et al., (2021) |
| | Hot impact on mortality | | | a = 3.1996 %K$^{-1}$<br><br>b = -0.0613 %K$^{-2}$ | |
| Labour productivity (Section 2.5.1) | Productivity of low-exposure labour | Quadratic in GMST anomaly (Form 1) | Not possible from literature. Internally calibrated in FRIDA | a = -2.0826 (-4.6663 - 0.5010)%K$^{-1}$<br><br>b = -1.5694 (-2 to -1.1387)%K$^{-2}$ | Dasgupta et al., (2021) |
| | Productivity of high-exposure labour | | | a = -6.133 (-7.823 to -4.442) %K$^{-1}$<br><br>b = -1.0620 (-1.5 to -0.6240) %K$^{-2}$ | |
| Indirect Economic | Scaling factor on default loan failure | Linear in GMST anomaly (Form 2) | Internally calibrated in | 0.5807 (0.35 - 0.85 )K$^{-1}$ | N/A |



| Effects (Section 2.5.2) | rate | | FRIDA | | |
|---|---|---|---|---|---|
| Government Spending (Section 2.5.3) | Climate -driven component of government consumption | Linear in GMST anomaly (Form 2) | Internally calibrated in FRIDA | 0.0706 (0.0543 - 0.0869) K$^{-1}$ | N/A |
| Sea Level Rise (Section 2.5.4) | See corresponding sections from Fig. 1 and Ramme et al. (submitted) for details. | | | | |
| Crop yield and plant growth (Section 2.6.1) | Crop yield | Linear in $CO_2$, quadratic in absolute GMST (Form 3) | Sampled Percentiles* | a = -265.39 %<br><br>b = 34.49 %K$^{-1}$<br><br>c = -1.222 %K$^{-2}$<br><br>d = 0.0553 %ppm$^{-1}$ | Franke et al., (2020) |
| Evapotranspiration (Section 2.6.2) | Scaling factor on baseline evapotranspiration rate | Linear in GMST anomaly (Form 2) | Internally calibrated in FRIDA | 0.06435 (0.02870 - 0.1) K$^{-1}$ | N/A |

**Table 1: Summary of the climate impact channels represented in FRIDAv2.1. * uncertainty analysis of these impact channels follows the "Sampled Percentiles" approach, discussed in the main text; only the median set of parameter values is therefore given in the table. See Fig. 1 for the modules which each impact channel targets in FRIDA, and Fig. 1 and Fig. A1 for graphs of the damage functions utilised. The Functional Form number refers to the equation in Appendix A.2.**






## 2.1 Energy

### 2.1.1 Energy supply

Many studies investigate the effect of climate changes on the energy supply of various sources (Yalew et al., 2020), but many focus on local responses, rather than larger-scale effects relevant to FRIDA. Additionally, many uncertainties persist on larger scales, with a literature review by Yalew et al., (2020) determining broad ranges in regional and global impacts for different sources of energy. This is unsurprising, due to the range of potential impact channels and scenarios available to be studied.

Since FRIDA models individual energy sources at the global level, climate change impacts were considered on this scale, for both renewable and fossil-fuel based energy production. As discussed in Section 2.6.1, impacts on crop production are addressed in FRIDA; this impact channel also affects bioenergy, and is accounted for in FRIDA as bioenergy utilises output from total crop production, which is impacted by this modelled effect. This follows the assumption that bioenergy crops are impacted on the global scale identically to food crops; this is unlikely, but in lieu of appropriate literature and to conserve

crop production within the energy and land use feedback processes, this approach was followed to estimate bioenergy climate impacts in FRIDA.

Estimated effects on solar and wind energy production are mixed in the literature, and generally small on larger spatial scales (Yalew et al., 2020). Changes in solar power can be driven by changes in irradiance driven by circulation and cloud shifts, and by temperature-dependent efficiencies. The global impact of these is uncertain and slight; Gernaat et al., (2021), when

analysing total solar potential in ISIMIP climate model output, find a general reduction in efficiency under climate change approximately balanced by an increase in irradiance over Europe and North America. Wind power capacity and potential depends on similarly uncertain changes in low-level winds, which substantially cancel on larger scales; Gernaat et al., (2021) find a slight global decrease in overall potential, though it is unclear how this might translate into actual capacity. Yalew et al., (2020) noted mixed overall reported impacts on wind energy systems. In light of the small magnitude and uncertain sign

of these effects, and especially with regionally-varying impacts partially cancelling out on larger scales, global climate impacts on solar and wind energy systems are not included in FRIDAv2.1.

Climate impacts on thermoelectric power production were found to be uncertain but consistently slightly negative on the global scale in the literature review by Yalew et al., (2020). One study in particular, Van Vliet et al., (2016), had a methodology consistent with the requirements for damage functions in FRIDA. Using a hydrological model to study the

effects of climate change on local streamflows and temperatures, coupled with an inventory of existing thermoelectric power plants, and a model for how the efficiency of thermal cooling depends on water temperature, Van Vliet et al., (2016) studied



the impact of these two effects – changes in streamflow and in water temperature – on existing thermoelectric energy production under future climate change using early ISIMIP climate data.

For implementation of this effect in FRIDA, the impacts on global thermoelectric production under both scenarios (RCP2.6
and RCP8.5) and decades (2020s and 2050s) studied by Van Vliet et al., (2016) were taken from their Table S5, and regressed against global mean temperatures using a quadratic fit with no intercept (see the Appendix Section A.3 for details on data processing and the functional form).

This reduction in thermoelectric efficiency applies to energy production in power plants cooled by freshwater. For incorporation into FRIDA, modifications due to both these conditions - energy generated in power plants, and freshwater
cooled - had to be implemented. First, the shares of energy from fossil fuels - split into coal, oil, and gas - which are generated in cooled power plants were calculated by dividing data on electricity generation by source (Ember, 2024) by total energy generation by source (Energy Institute, 2024) with the implied assumption that electricity generation from fossil fuels (as opposed to heating or powering mobility) occurs in thermal power plants. For nuclear power, it was assumed that all energy production occurs in cooled power plants. Then, the share of power plants cooled by freshwater, described as 70.2%
by Van Vliet et al., (2016), was calculated. This factor was assumed to be independent of the specific fuel. This contribution to energy production was then modified by the efficiency change due to climate change.

These represented effects are driven by long-term changes in water temperature and streamflow - therefore they indicate the longer-timescale response, and cannot be interpreted as incorporating the impact of higher temporal-resolution processes such as storms.

As well as studying climate effects on thermoelectric power, Van Vliet et al., (2016) investigate the impact of climate change on hydropower production. Unlike thermoelectric plants, hydropower plant efficiency is unaffected by water temperature; only the effects of changes in streamflow are modelled. Using data from their Table S4, and the same GMST data as used for the thermoelectric supply, a zero-intercept quadratic fit was applied. These impacts are lower than for thermal power generation, due to the lack of a water temperature effect. Since this effect applies to the hydropower sector as a whole, the
further splicing of energy production performed in the thermal plant case was not needed here.

That the net effect of streamflow changes is negative reflects the overall effect of increases in very high latitudes and decreases in e.g. southern Europe and South America (Van Vliet et al., 2016). The net effect therefore depends on the distribution of power plants, which can be expected to change in the future due to multiple factors, including adaptation. No distributional changes are incorporated into the representation of this damage function in FRIDAv2.1.

The effects modelled by Van Vliet et al., (2016) represent only these specific impact channels – changes in streamflow and water temperature – and do not address other channels such as the effect of extreme events on infrastructure. An additional channel in FRIDA is designed to account for these infrastructure damages, presented in Section 2.1.3.



**2.1.2 Energy Demand**

Under a warming climate, energy use for air temperature control can be expected to change. Potentially a key system feedback, due to the close link between energy systems and emissions, the net effect is the balance between a reduction in heating demand and an increase in cooling demand under global warming (Kennard et al., 2022; Rode et al., 2021; Yalew et al., 2020). Since these two components of energy demand change generally feature different energy carriers – electricity-powered air conditioning for cooling, and (typically) the burning of fossil fuels for direct heating – their drivers vary, with

their net effect dependent on the complex balance of differing processes, particularly the level of air conditioning penetration (Clarke et al., 2018; Rode et al., 2021; Yalew et al., 2020). There is an overall spatial pattern to the response, with general decreases in heating in high latitudes, and increases in cooling nearer the equator, with a consequent dependence on the projected pattern of climate change (Deroubaix et al., 2021; Hartin et al., 2021). The effect on energy demand is further modulated by the population distribution (Kennard et al., 2022).

Since FRIDA aims to explain feedback loops within the coupled human-Earth system using meaningful concepts at the process level, the effect of climate change on energy demand has been implemented in a two-step process. First, a link between the level of global warming and the change in energy-relevant climate metrics was established. This was then linked to the overall change in demand.

  The energy-relevant climate concept utilised in FRIDA is the widely used Cooling and Heating degree days (CDDs and

HDDs). These represent the total exceedance of local temperatures of given hot and cold thresholds, summed over the daily differences between the threshold and the actual temperature. Many implementations of this concept use simple summation of exceedances over various thresholds, with typical heating and cooling thresholds of 18°C and 22°C respectively (Kennard et al., 2022; Miranda et al., 2023). A more complex method defined by the UK Met Office involves utilising maximum and minimum temperatures in addition to the mean, allowing for heating and cooling on the same day (Deroubaix et al., 2021;

Spinoni et al., 2021). CDD and HDD are used as scaling proxies for heating demand, since they approximately represent the level of cooling and heating required on a given day. However, their lack of accounting for humidity renders CDD and HDD a suboptimal framework for analysing heating demand (Kennard et al., 2022).

  HDD and CDD have undergone decreases and increases respectively in observations since 1981 (Kennard et al., 2022), with their relative magnitude dependent on the choice of threshold, though historical trends are weak compared to future

projections (Deroubaix et al., 2021). Absolute HDD and CDD responses to climate change feature the expected spatial structure – reductions in the former centred over high latitudes and increases in the tropics in the latter (Deroubaix et al., 2021) – though in relative terms the change in CDD is largest in high latitude areas (Miranda et al., 2023).

  Arnell et al., (2019), using CMIP5 model data, found approximately linear global changes in CDD and HDD with global temperatures, weighted over populated regions. Using regional CMIP5 data coupled to SSP population data, Spinoni et al.,





(2021) found substantial changes in population-scaled CDD and HDD, with strong socioeconomic dependence at given
global warming levels due to the population scaling.

For FRIDA, population-weighted CDD and HDD are required, to capture the energy demand-driving component of change.
Adapting the framework by Werning, Hooke, et al., (2024), in FRIDA the response of global average annual CDD and HDD
per person, using thresholds of 22°C and 18°C respectively, under a range of SSP projections of emissions and population,
was calculated and modelled as a function of GMST.

Based on the fits and their residuals, for implementation in FRIDA, CDD was taken to increase approximately linearly in
GMST, while HDD declines with an exponential decay (see Appendix Section A.4 and Fig. A2 for the fits and their
functional forms, and further data details).

These projections of population-weighted CDD and HDD were then combined with literature estimates of the overall impact
of climate change on heating and cooling energy use, to represent the full effect. Clarke et al., (2018) find a net increase in
residential energy use, with increased cooling outweighing decreased heating; however Rode et al., (2021) find the opposite
effect, with a net decrease in energy use. Varying assumptions around access to air conditioning for cooling determine the
overall balance in demand. In FRIDA, the sensitivity of demand to changes in CDD and HDD are separately calibrated to
both these estimates, in order to explore the uncertainty within this impact. If FRIDA were to sectoralise in future
development, the analysis by van Ruijven et al., (2019) could be utilised to drive sector-specific responses. Within FRIDA,
the change in CDD is then also directly linked to consequent emissions of HFCs required for air conditioning via the
HFC134a-equivalent aggregated basket species for HFC emissions, since this sector is a key source of fluorinated gas
emissions (Hartin et al., 2021).

The uncertainty in the CDD and HDD functions could have been explored within the data (see Fig. A3), but instead, a
different approach was taken, whereby the parameters were only narrowly varied as part of the internal calibration (see Table
1 and Fig. A1). This was done because of the direct linking of CDD and HDD to literature-based energy demand responses,
with their own internal uncertainty. The uncertainty in the CDD connection to HFC emissions is also internally calibrated.

These literature-based responses reflect the process-level representation contained within the available literature; they can
conceptually be thought of as driven by temperature-related shifts in long-term means and shorter-term processes such as
330   heatwaves.

Adaptation measures, such as changes to building standards reducing future air-conditioning demand per CDD, are not
included mechanistically within FRIDAv2.1. Within the policy levers associated with Behavioural Choices, individual
energy demand can be externally driven to explore the effect of demand-linked policy implementations.

### 2.1.3 Energy Infrastructure Damage

335   As detailed in Section 2.1.1, the effect of climate change on the efficiency of hydro- and thermo-electric power plants is
represented in FRIDA, based on a literature estimate. However, no suitable literature to represent the global effect of climate
change on energy systems infrastructure was identified. This impact channel was therefore incorporated via an internally



calibrated parameter (see Schoenberg et al., submitted), with damages on energy infrastructure represented as a power law function of global temperature anomaly. This functional form was chosen based on its prevalence in IAM literature (Füssel, 2010; Nordhaus, 2014). The parameters of the function are specified with a large uncertainty range reflecting the lack of knowledge to constrain them. The decay rate of energy capital stocks due to climate impacts is then given as:

$$D = a\Delta T^b.$$

(1)

Here $T$ is the global mean temperature anomaly, and $D$ is the decay rate of energy infrastructure due to climate change. In FRIDAv2.1, the decay rate is applied to the capital stocks of all energy sources equally.

## 2.2 Behavioural Change

### 2.2.1 Exposure to Climate Extremes

Populations' exposure to climate change-driven extreme events is expected to alter risk perceptions and thus behaviour, including those which influence anthropogenic climate forcing, which will alter the further evolution of the human-Earth system (Beckage et al., 2022; van der Linden, 2015). This feedback loop, initiated by the climate response, is therefore classified as a climate feedback within FRIDA. Here, the connection from global temperature change to human exposure to extreme climatic events is focused on, while the downstream adaptive behavioural processes in, and response to, climate change risk perceptions is expanded upon in (Rajah et al., 2025).

Under historical climate change, exposure to climatic extremes has already been altered, and will increase at different rates under a range of scenarios. Lange et al., (2020) used ISIMIP2b data to suggest exposure to various climate impact events has increased 2.4-fold since pre-industrial times, with substantial further increases projected through the 21st century. Looking specifically at annual mean temperatures, Lenton et al., (2023) found that the population exposed to temperatures above 29 °C have tripled to 1% since 1980, with this metric increasing to 8-40% by 2100. Changes in exposure such as these are expected to drive subsequent shifts in people's behaviour, representing mitigation of and/or adaptation to climate changes (Beckage et al., 2018, 2022).

Myriad indicators and exposure metrics are used in the literature, covering a range from multiple sectors (Lange et al., 2020) to purely heat-based exposures (Schwingshackl et al., 2021). In psychological models, personal experience with extreme weather events has been found to be a significant predictor of climate change risk perception (Akerlof et al., 2013; Xie et al., 2019). Therefore, FRIDA uses individual population-average exposure to local weather extremes to partly drive the experiential process in risk perception. This population-weighted measure will vary under future scenarios due to changes in both climate and population distribution.

While Schwingshackl et al., (2021) have found a range of heat stress indicators to scale approximately linearly with GMST, with the gradient dependent on the chosen indicator, they only cover temperature (and humidity)-based metrics. Instead, B.



Li et al., (2023) studied the change in population-scaled exceedances of historically-observed extremes in downscaled CMIP6 data from 35 models for the four "Tier 1" SSPs (SSP1-2.6, SSP2-4.5, SSP3-7.0 and SSP5-8.5), across a range of indices relating to temperature, precipitation, and compound events. A person was counted as exposed if they experienced an event more extreme than the historically-observed events for that gridcell. This was presented as decadal averages from the 2020s to the 2090s across a range of scenarios. For use in FRIDA, this was normalised by the global population, converting

to the population-averaged response, and then summed over the seven available indices (Annual total precipitation, Maximum 1-day precipitation, Number of days with heavy precipitation, Warm days, Heatwave, Sequential precipitation-humid heatwave, and Compound drought and heatwave).

The overall metric therefore represents the average number of indices of which a person experiences at least one weather event in a given year that is more extreme than the historical record. This total exposure to extreme weather events was then

expressed as a zero-intercept quadratic function of GMST using the corresponding global temperatures (see the Appendix Section A.5 for the fit, and Fig. A3 for further details on the chosen indices). The imposed zero intercept shifts the exposure baseline, rendering the metric less directly measured, but this was necessary to represent exposure relative to pre-industrial levels. Through the incorporation of these metrics, the implied global exposure to heatwaves and storms, as well as changes in mean precipitation and compound events, are accounted for in FRIDA.

In FRIDA, exposure to sea level rise (SLR) is differentiated from temperature-related extreme weather events for climate change risk perception. SLR impacts are not only felt by comparatively smaller coastal populations, after additionally accounting for resilience and previous retreat, but also experienced as storm-surge flooding, which people may not directly associate with global climate change (Akerlof et al., 2017). Accordingly, SLR exposure is not included in the weather-related indices described above. Instead, exposure to flooding from SLR is included as a distinct risk perception process.

FRIDA tracks the number of people in the coastal zone flooded annually from SLR-related storm surges (Ramme et al., submitted). Future increased exposure to SLR-induced flooding is expected to result in increased public attention (Akerlof et al., 2017). Consequently, risk from SLR is perceived by the broader global population only when this metric crosses a large enough threshold, assuming it would receive international media attention and act as a vicarious experience (Akerlof et al., 2013).

**2.3 Resources**

**2.3.1 Durability of Concrete**

Concrete is the primary material used in modern housing and service building construction (Marinova et al., 2020) and constitutes a significant share of material used in infrastructure (Deetman et al., 2020). Its production contributes around 8-9% of anthropogenic $CO_2$ emissions, via emissions from cement, and approximately 2-3% of global energy use (Monteiro et

al., 2017). Any feedback between climate change and the use of concrete may therefore represent an important feedback within the coupled human-Earth system.





In FRIDA the global concrete sector is represented in a highly aggregated form consisting of two types of use: concrete in housing and service buildings, and concrete in infrastructure. Concrete structures of both types are built and as they age, they move from the stock of new structures to the stock of old structures, before leaving the system upon reaching their service lifetime. Climate impacts are represented according to the logic that higher global average temperatures reduce the durability of concrete structures through carbonation- and chloride-induced corrosion, ultimately resulting in damages to the rebar structures of the buildings. Both types of corrosion are caused by a combination of temperature, $CO_2$ and chloride-exposure mechanisms (Bastidas-Arteaga and Stewart, 2015; Stewart et al., 2011). The magnitudes of these impacts are highly context-dependent and vary with geography, distance to the ocean, type of structure, concrete mix, etc. Global-level averages for the impact are not available from the literature.

The service lifetime of concrete structures is therefore represented as declining by 5-15% linearly for every degree Celsius increase in global average temperature using the base average service lifetime of the structure (a calibrated parameter) as a baseline. In FRIDA a best estimate of 7.5% lifetime decline per degree is used, based on estimates of increased corrosion rates. The range was derived from qualitative evaluation of existing literature of local-to-regional scale climate impacts on reinforced concrete durability (Bastidas-Arteaga and Stewart, 2015; Stewart et al., 2011; Wang et al., 2012). This reduction in service lifetime of concrete structures as a consequence of climate change, mediated by enhanced corrosion, is driven by three processes. First, elevated atmospheric carbon concentration accelerates carbonation and increases the carbonation depth in concrete, which increases the risk for corrosion and structural damage to the rebar structure. Second, elevated ambient temperatures accelerate carbonation and chloride penetration, which exacerbates corrosion initiation and the risk of structural damages. Third, changes in relative humidity due to higher temperatures may allow carbonation and chloride penetration in regions where these damages previously have been of minor concern (Wang et al., 2012). Due to the simplistic nature of the representation in FRIDAv2.1, these impacts are currently driven only by global temperatures; future development could incorporate $CO_2$ concentrations, moving towards a process-based representation of the corrosion effects.

While techniques for adaptation to this climate impact are not currently simulated in FRIDA, the process-level representation allows for the possibility of altering parameters to address this, such as by increasing building lifetimes, adjusting building standards, or changing the concrete mix to more durable alternatives. Cost considerations such as investments in research and the price differential of more durable alternatives would be explored under these possible adaptation methods.

To account for SLR damages on concrete use, the annual SLR-driven storm surge damage to coastal assets is multiplied by the average historical concrete per asset ratio. This represents a simple initial implementation for the linkage between SLR and concrete resources, which can be expanded upon in future development.



## 2.4 Demographics

### 2.4.1 Mortality

Extreme hot and cold temperatures are associated with increased mortality (Cromar et al., 2022). These associations are significantly stronger for older populations (Chen et al., 2024), and operate through complex mechanisms (Masuda et al.,
435 2024).

Discrete observations of mortality from specific, extreme events are substantially hindered by data availability issues, with strong geographic inequalities (Newman and Noy, 2023). While impacts of climate change on human mortality can be expressed in terms of economic damages (Newman and Noy, 2023), in FRIDAv2.1 the approach is to directly include the mortality effect, with subsequent effects on the other human and climate systems occurring as a result.

A meta-analysis of epidemiological studies which modelled responses to hot and cold extremes produced regional risk functions, finding that increased heat-driven deaths under climate change consistently outweighed decreases in cold deaths on the regional level (Cromar et al., 2022). For FRIDA, the empirical analysis presented by Bressler et al., (2021) was utilised to project global impacts of climate on mortality. Using historical data, Bressler et al., (2021) built links between daily mortality and daily mean temperature, separately for hot and cold impacts. While they express results on the country
level, additionally accounting for income changes, they present global impacts as a function of GMST in their Fig. 3. This data was traced, and a zero-intercept quadratic fitted. The temperature data was offset to be relative to pre-industrial levels, thereby expressing the impacts as relative to the pre-industrial level. As noted in Section 1, it was decided to express uncertainties within climate impacts within 5-95 percentiles, where possible, to maximise consistency. The data in Bressler et al., (2021) is given for these 5-95 percentiles, allowing their uncertainty representation to be directly used. The functional
form and further data details are given in the Appendix Section A.6.

The use of daily data implies the incorporation of the effect of events on this timescale; i.e. that short-term heatwaves are included as drivers of this mortality effect.

Investigating the effect of ageing on temperature-driven mortality, Chen et al., (2024) presented scaling factors for mortality rates between different age groups; in the supplementary material they note that "Compared with the 0-64 age group, the 65-
74 and ≥75 years age groups have 1.2 and 1.8 times of heat-related mortality risks and 1.9 and 2.4 times of cold-related mortality risks". In FRIDAv2.1, this information was combined with the population fractions in 2024 to assign different mortality rates to these three age bands.

The approach taken in FRIDA therefore allows for the accounting of temperature mortality, split between hot and cold effects and by three age brackets. These health impacts are modelled at the global level but have a pronounced regional
dependence, which may be expected to drive different dynamics than a globally uniform effect. However, Cromar et al., (2022) found that increased heat deaths outweighed decreased cold deaths in each region, suggesting a consistent overall dynamic, albeit varying significantly in strength. While it is plausible that measures could be taken to reduce heat exposure and thereby partially adapt to this climate damage, it is unclear how best to simulate this at the process level within a model



such as FRIDA, or that this incorporation would substantially alter the trajectory of the overall system; investigation of this

will form part of the future development of the model (Section 3.3).

Similarly to how FRIDA calculates the increasing exposure of coastal assets, it also calculates the increasing exposure of the coastal population to SLR-driven storm surge damages using a logistic function (Ramme et al submitted). After accounting for resilience and previous retreat of the coastal population, the fatality rate of the people that are living in flooded areas in one year is assumed to be 1%, as applied in Wong et al., (2022), adding an assumed uncertainty range of 0.5-2%. The loss of

life is then added to the death rates of the individual age groups within FRIDA.

## 2.5 Economy

### 2.5.1 Labour Productivity

The ability of people to carry out activities such as work depends on the climate they experience. While effects of hot and cold temperatures affect overall work levels (Dasgupta et al., 2021), combinations of high temperature and humidity can

make ambient conditions unliveable, when the wet bulb globe temperature (WBGT) exceeds uncertain thresholds (Clarke et al., 2022). This occurs via the limitation of the ability of humans to cool down via sweating, as well as by enhancing the risk of heat stroke (Masuda et al., 2024).

While it is known that physiological reasons cause limits to work above certain WBGT levels, the value of this limit is uncertain between laboratory and empirical studies (Masuda et al., 2024; Orlov et al., 2020), partly due to uncertainties

around adaptation practices, which may be numerous and complex (Masuda et al., 2024).

Reductions in work levels due to uncomfortable heat levels occur substantially below any physiological WBGT limit, and vary between regions due to historical adaptation (Dasgupta et al., 2021), raising the complex question of future adaptations under climate change. These impacts depend on the nature of the work in question, with impacts split by work intensity and/or weather exposure (Dasgupta et al., 2021; Orlov et al., 2020).

Substantial aggregate climate impacts have been attributed to reductions in worker efficiency under climate change. Orlov et al., (2020) found a 1.4% drop in GDP in 2100 under the high-emissions RCP8.5 scenario; in a follow-up study focused on the agricultural sector, Orlov et al., (2021), using the GRACE computable general equilibrium model, found considerable regional food price inflation due to increased production costs.

For FRIDAv2.1, the results of Dasgupta et al., (2021) were utilised to model the impacts of climate change on labour output.

Dasgupta et al., (2021) used empirical survey data to study the effect of climate on hours worked – termed "labour supply" – and coupled this with prior research on the impact of climate on the effectiveness of work done – "labour productivity" – to get the overall change in "effective labour". This effective labour then includes both the effect of changed working hours and changed work efficiency under climate change. Dasgupta et al., (2021) split work into two levels of exposure: High, designated as "outdoor work in the sun", and Low, defined as "indoor work or outdoor work in the shade".





The use of empirical survey data by Dasgupta et al., (2021) reduces concerns on the discrepancy between idealised laboratory studies and real-world impacts highlighted elsewhere (Masuda et al., 2024). One concern, however, is that the recent (1986-2005) estimated impacts of climate on labour are substantial, especially in the tropics, with widespread regions with around 60% drops in work levels compared to the optimum climate (see their Fig. 3).

For use in FRIDA, data from Dasgupta et al., (2021) for the combined global effect of climate change on effective labour at
1.5°, 2°, and 3°C of global warming, relative to pre-industrial levels, were taken, and a quadratic function in GMST anomaly fit to the data, with an imposed zero intercept by definition. Note that, since uncertainty estimates were not taken from Dasgupta et al., (2021), the uncertainty bounds were determined from internal calibration within FRIDA (see the Appendix Section A.7 for further details and the functional form).

It should be noted that future changes in the spatial distribution of sectoral work would affect this aggregate impact; for
example, if high-exposure manufacturing work shifted to more climate-exposed regions, the overall climate impact would be enhanced. This is considered a second-order effect currently, and is therefore ignored in FRIDAv2.1, though there may be scope to include this effect in future (Leimbach et al., 2023). In addition, in FRIDAv2.1 no explicit representation of adaptation to this climate impact is modelled; in future, investment-driven adaptation could plausibly increase the share of low exposure labour relative to high exposure labour, partially mitigating this effect.

Since Dasgupta et al., (2021)'s results are recorded weekly, with significantly different results based on daily temperature, it can be assumed that these global temperature-driven effects are ultimately mediated by short-term temperature-based drivers, such as heatwaves.

In addition to the GMST-driven response, SLR-induced flooding adversely impacts labour productivity. Using the number of people that are living in flooded areas annually, accounting for resilience and previous retreat, the approach taken by the
FUND model is followed, by which it is assumed that people in flooded areas do not work for two weeks (Tol, 2007; Waldhoff et al., 2014). This translates into a reduction factor of 0.96 for their annual productivity, which is expanded with an uncertainty range between 0.9 and 0.99. The overall productivity reduction because of SLR then becomes $1 - \frac{\text{People flooded}}{\text{Global population}} \text{reductionFactor}$. As this is a relative effect on productivity, the unemployment rate and working age population fraction are accounted for.

**2.5.2 Indirect Economic Effects**

The climate impacts described by the other damage functions in FRIDAv2.1 cover direct physical effects of economic relevance. However, they do not cover the entire spectrum of potential economic consequences of climate change. A growing body of evidence and scientific literature points to the fact that the most significant impact of climate change is likely to be economic in nature, manifesting as shifts in expectations as climate uncertainty changes into climate risks, and is
therefore priced by markets (Kantor, 1979; Stroebel and Wurgler, 2021), leading to cascading negative impacts in financial assets evaluation, with liquidity and solvency consequences (Bartsch et al., 2025; Carattini et al., 2023; Feng et al., 2024).



These effects are likely to spread well beyond the range of physical damages, affecting the evaluation of both present and future investments alike. As extreme climate events increase in both number and scope, economic assets that are physically whole yet potentially at risk of being damaged in a future event are also diminished in value, as expectations and market prices adjust to reflect the new information.

The complex nature of these indirect economic effects would require detailed financial and monetary markets to be satisfactorily explored. While FRIDA includes financial and monetary mechanisms, their highly aggregate nature limits the representation of broader economic damages to an aggregate depiction of these effects, using the failure rate of loan as the primary process. In FRIDA, investments are associated with a failure rate, which rises linearly with GMST (Fig. 1), leading to increased, unexpected bankruptcies and associated non-performing loans. These unexpected defaults produce negative employment shocks in the short term and long-term changes in lending standards. As the financial sector revises future growth expectations downwards to account for climate damages, the social costs of climate change are mitigated but investment, and therefore economic development, is curtailed (Dell'ariccia and Marquez, 2006; Fishman et al., 2024; Lown and Morgan, 2006; Rodano et al., 2018).

The strengths of these indirect economic effects are calibrated as part of the full-model calibration, using past data. This implies that these indirect economic effects are at work for the entire range of the model simulation. It might be argued that small temperature changes are not generally associated with significant direct climate impacts. Indirect economic effects, however, are primarily mediated by expectations; as evidence for future climate risks increase, reducing uncertainty, their weight in economic calculation similarly increases. Consequently, while logically secondary to other climate damages, the compounding, forward-looking nature of these effects implies that their economic consequences are significant.

### 2.5.3 Government Spending

Climate change significantly affects public spending by increasing costs for infrastructure repair, adaptation, and disaster response. Extreme weather events like floods, hurricanes, and wildfires can be expected to damage critical infrastructure such as roads, bridges, power grids, and water systems, leading to costly repairs. These expenses strain public budgets, potentially reducing government investments and therefore hindering long-term economic growth (Avtar et al., 2023). In FRIDA, this process is simulated using the global mean surface temperature anomaly to linearly increase the fraction of government spending allocated to consumption (which includes repairs) at the expense of government investment. This extra climate-driven government consumption increases the debt to GDP ratio in the long-term, which over time compounds by increasing government interest payments. Both of these impacts cause there to be less money available for investment by the government. The impact of this structure on overall economic growth is complex, as the lack of investment pulls down GDP, but the extra government spending tends to increase it, provided that economic resources are not already fully utilized.



### 2.5.4 Sea Level Rise Effects

There are three distinct SLR-driven economic impact channels in FRIDAv2.1 not discussed elsewhere in this section. These
affect the loan failure rate - which is also connected to GMST (Section 2.5.2) - as well as additional financial cost-based
channels.

Firstly, SLR damages coastal assets. The major component of this is the increase in the severity of storm surges, as SLR
raises the base level of the surge (Cazenave and Cozannet, 2014; Wong et al., 2022). This is reflected by modelling the
annual exposure of coastal assets to storm surges using a logistic function of the mean sea level anomaly (subtracting any
rise in the flood protection height through protection investments), modified by the resilience of the coastal zone and
previous asset retreat (Ramme et al., submitted). The other component of damage to coastal assets is the loss of assets due to
inundation or retreat. This affects only the immobile part of assets, and, in the case of proactive retreat, 80-100% of the
assets' value is assumed to be depreciated at the time of retreat, therefore not causing economic damage. The loss of assets is
linked to the financial module of FRIDA and interpreted as a loss of safe loans. Therefore, SLR-driven asset damages affect
a similar part of the economy as the temperature-driven increase in the failure rate of loans (see Section 2.5.2), but they are
applied conceptually differently. While the temperature effect is modelled as a change in the non-performing fraction of
newly issued loans, the SLR effect is assumed to represent the defaulting of loans that would have otherwise been safe.
However, the downstream economic consequences (unemployment shocks and a tightening of lending standards) are the
same.

Secondly, people and assets can retreat from the coast because of SLR. This can either happen as an adaptation strategy to
avoid future damages from storm surges, or it can be forced, when a home or an asset becomes inundated. Retreat causes
costs for moving mobile assets, costs to demolish abandoned immobile assets and costs to relocate people. A full description
of how those costs are defined can be found in Ramme et al., (submitted). In FRIDA, those costs are summed and interpreted
as owner consumption in the economy module of FRIDA. This is done because it is the owners of assets that must pay for its
relocation. Owner consumption in FRIDA transfers money from owner savings to firms' checking accounts.

Finally, instead of retreating or conducting no adaptation at all, the coastal zone also has the option to protect against future
SLR-driven damages and inundation. The chosen formulation translates flood protection investments into an increase of the
height of a generic seawall, whereby the costs to raise and maintain the flood protection grow quadratically with height
(Wong et al., 2022; Ramme et al., submitted). The investments into raising the flood protection height, as well as the
additional maintenance costs of the raised flood protection, are assumed to be unproductive government expenditure in
FRIDA. This means that flood protection costs are added to public expenditure, but they are not accounted for in the part of
public expenditure that increases investments and consumption and thereby can grow the economy. The increase in
unproductive public expenditure leads to the government taking up more debt and increases its debt to GDP ratio, which can
impact future government expenditure on other expenses.



## 2.6 Land Use & Agriculture

Several connections are made between the Climate module and the Land Use & Agriculture module in FRIDAv2.1. Two of these - representing impacts on the agricultural sector, via crop yields and on irrigation evapotranspiration – are clear climate impact channels, imposing on domains within the human system, in this case agriculture, and are detailed here. The other connections, while still relevant for human systems and connected to them within FRIDA, are instead considered internal climate feedbacks, linking across modules in FRIDA due to the necessary split of the carbon cycle between these modules. These connections are explored in Section 7.

Many methods have been developed to investigate the impacts of climate change on the agricultural sector. As when measuring whole-economy impacts, these can be separated into top-down and bottom-up approaches (Piontek et al., 2021), with FRIDA implementing a process-based, bottom-up approach. Broader sectoral impacts extend to the impact of climate change on workers within the agricultural sector (Orlov et al., 2020), which is incorporated in FRIDA as described in Section 2.5.1, and changes to the efficiency of bioenergy infrastructure (Gernaat et al., 2021), with the approach in FRIDA detailed in Section 2.1.1. Cascading impacts from the agricultural response can raise the social cost of carbon (Moore et al., 2017), with supply-demand balance shifts causing effects on food prices (Kompas et al., 2024; Orlov et al., 2021; Rezaei et al., 2023), which manifest in FRIDA through the Economy module. Projected rainfall changes may also increase income inequality through impacts on the agricultural sector (Palagi et al., 2022), though this level of detail is not represented in FRIDA.

Changes in the agricultural sector cause associated feedbacks onto the climate system not just from greenhouse gas emissions, but also via changes in surface albedo (Ghimire et al., 2014) and $H_2O$ emissions from irrigation (Sherwood et al., 2018); these effects are simulated and passed to the FRIDA Climate module. While these effects are slight, their inclusion incorporates interactions between the climate and human system only possible in a coupled model such as FRIDA.

### 2.6.1 Crop Yield

The approach in FRIDA focuses on a key variable in the food systems sector, namely crop yield. FRIDA represents only a single, aggregated crop yield, with no differentiation of different crop types. Changes are therefore represented in FRIDA in this crop- and region-aggregated yield, as a function of climate variables.

Effects of climate change on yields have been explored via a range of approaches, from localised field experiments, to process-based modelling, to empirical statistical modelling (Hu et al., 2024; Zhao et al., 2017). Complex, nonlinear responses varying between regions and crops are reported, with a strong dependence on the method or model used (Hu et al., 2024; Müller et al., 2021; Ruane et al., 2024). Crop yields are affected by multiple climate drivers; temperature, $CO_2$, and precipitation. The effect of increased $CO_2$ is to increase yields, via the $CO_2$ fertilisation effect, with a strong dependence on crop type and water stress (Ostberg et al., 2018), and uncertain magnitude (Gernaat et al., 2021; Müller et al., 2021). This



$CO_2$ effect precludes the possibility of using global mean temperatures in FRIDA as the sole driver of the climate impact on crop yields (Ruane et al., 2024), as applied for some other impact channels in FRIDA.

Early single-model studies determined slight changes in crop yield under climate change (Levis et al., 2018; Tebaldi and Lobell, 2018), with the $CO_2$ fertilisation effect a key source of uncertainty (Ren et al., 2018). The ISIMIP project seeks to
address these uncertainties using many experiments in a multi-model comparison (Müller et al., 2019; Ostberg et al., 2018). ISIMIP crop models generally simulate present-day yields well (Franke et al., 2020b; Müller et al., 2024), but models still exhibit large differences under future climate conditions (Heinicke et al., 2022; Müller et al., 2024).

In a recent literature review, Hu et al., (2024) found an overall decrease in yields of key crops under 1°C of global warming, though they note many studies neglect the $CO_2$ fertilisation effect. In the most recent round of the ISIMIP project,
projections of uncertain yield changes reflect the complex balance between the responses to $CO_2$ and temperature (Müller et al., 2021). The global mean response in crop yields masks the spatial pattern of the response, which typically includes increased yields in high latitude regions and decreased yields in lower latitudes (Franke et al., 2020b; Ostberg et al., 2018), driven by the high baseline temperatures and predominance of less $CO_2$-sensitive C4 crops in the tropics (Rezaei et al., 2023).

Many studies focus on four main crops – wheat, maize, soybean, and rice – which comprise the majority of produced calories (Franke et al., 2020b; Hu et al., 2024; Müller et al., 2019; Zhao et al., 2017), with wheat further split into summer and spring production in the latest round of ISIMIP simulations (Franke et al., 2020a). The aggregation of these crops via calorie-weighting reduces overall inter-model uncertainty via compensating differences (Müller et al., 2021).

For use in FRIDA, data from analysis of the ISIMIP project was used (Müller et al., 2021). The experiments in ISIMIP input
transient, realistic future climate scenarios (the SSPs) into crop models to study their effect. This only explores a small region of the driver space, since in many scenarios there is a strong correlation between temperature and $CO_2$. To address this, a "parameter sweep" approach has been carried out using ISIMIP models, whereby the driving dimensions of temperature, $CO_2$, water, and nitrogen are systematically perturbed (Franke et al., 2020a). This allows for the creation of grid-cell, crop, and model-specific functions with which to emulate the response to an arbitrary change in these drivers,
accounting for their interaction effects (Franke et al., 2020b). This allows for further exploration of inter-model differences, with Müller et al., (2024) finding different relative importance attributed to different drivers between models. These derived functions, which accurately reproduce the individual model output (Franke et al., 2020a), have then been applied to transient scenario climate data from CMIP5 and CMIP6 (Müller et al., 2021), in order to more fully explore the scenario and climate model space.

This data from Müller et al., (2021) was used to generate the impact function for FRIDA. They estimated the response of crops aggregated over the four main cultivars (wheat, maize, soybean, and rice) and accounting for the current irrigation regime to estimate changes in total crop production in CMIP6 models under the SSPs, as well as idealised scenarios with zero $CO_2$ response in order to better explore the parameter space.




The sum over these key crop types was taken to match the single aggregate crop type in FRIDA. Changes in this total crop
production were modelled as linear in $CO_2$ concentration, with first and second order temperature-mediated effects (see
Table 1 for parameters, and the Appendix Section A.8 and Fig. A4 for the functional form equation and figure). Two-
dimensional slices of the optimal fit against GMST and $CO_2$ are shown in Fig. 1.

Current crop models are thought to underestimate negative climate impacts on yield, from droughts and heatwaves (Heinicke
et al., 2022) to extreme high rainfall (Liu et al., 2023), which has been found to negatively impact the agriculture sector in
poorer regions (Kotz et al., 2022). In addition, potential impact channels such as changes in the levels and location of pests
and diseases are not included in current crop models (Franke et al., 2020b), and therefore not reflected in FRIDA.

FRIDA represents the processes which close the gap between (crop) supply and demand. This is achieved via increased land
expansion, fertiliser use, and irrigation. The latter two include technologically-driven improvements over time, which
conceptually incorporate simple adaptations to this climate response under climate change.

Impacts of global temperatures on crops can occur through several mechanisms, from changes in plant growth rates, to
flooding, heatwaves, and storms. To the extent that these mechanisms are represented in the crop models used to produce the
utilised data, these physical mechanisms are accounted for within the FRIDA model.

### 2.6.2 Evapotranspiration

In FRIDA the impact of increasing surface temperatures on water lost to evapotranspiration during irrigation is represented
on a per irrigated hectare per year basis. This is modelled using a calibrated, shallow linear function based on the GMST
anomaly. Counteracting this effect is a representation of technological development, and the further rollout of more efficient
irrigation methods. This technological process is represented using a calibrated negative exponential function which reduces
the amount of water applied per irrigated hectare per year, which reduces future evapotranspiration losses. Potential
evaporation is primarily affected by surface air temperature, atmospheric humidity, wind speed, and net radiation (Monteith,
1965; Allen et al., 1998), and their respective importance with context. Temperature is often the most important variable (see
e.g., Guo et al., 2017; Shi et al., 2020), and temperature-based equations are often used when data on other climate variables
are not available (H. Hargreaves and A. Samani, 1985; Hamon, 1963). The relationship between air temperature and
potential evaporation is not necessarily straightforward, due to e.g., soil moisture limits or negative atmospheric humidity
feedback (increased temperature with higher humidity, which reduces evaporative demand).

The assumed linear relationship between air temperature and water withdrawal in FRIDA likely captures the overall
tendencies, but in future development the extent of the validity could be checked against spatially distributed global models,
as FRIDA doesn't explicitly represent changes in the patterns of climate variables under climate change which may affect
potential evaporation. This includes wind speed (McVicar et al., 2012; though wind pattern changes under climate change
are highly uncertain, as noted in Section 2.1.1), and precipitation and plant-available soil moisture, which may affect crop
water demand from irrigation. This function is therefore internally calibrated.





## 3 Discussion

### 3.1 The present and future scope of climate impact channels in the FRIDA IAM

As discussed in Section 2, the set of climate impact channels explored in the Technical Summary to the WGII report of the
IPCC AR6 (Pörtner et al., 2022) were split into three general types during the process of implementing damages in
FRIDAv2.1: (1) impact channels deemed essential and were possible to include in FRIDAv2.1, (2) impact channels to be
considered for future development, and (3) impact channels that are likely impossible to represent in a globally aggregated
type of model.

The first category is documented in Section 2 and summarised in Table 1 and Figs. 1 and A1. These were deemed essential
to be represented within FRIDAv2.1 to provide a reasonably comprehensive set of climate damages, in such a model as
FRIDA, to be able to explore the future trajectory of the coupled human-Earth system. The implementation of this broad set
of climate damages, at the process level, allows for the exploration of the complex and cascading impacts of climate change
on the coupled human-Earth system in FRIDAv2.1, in the context of its broader feedbacks and system interactions, and with
a comprehensive investigation of the uncertainties in the responses.

The second category comprises those impacts not currently included within FRIDA, but which could be explored as further
development. For example, fires affect the carbon cycle and the climate, as well as interacting directly with human systems,
although the magnitude of their effect on populations is relatively small and driven also by complex socioeconomic
dynamics (Clarke et al., 2022; Park et al., 2024). Their representation via the approximation of burnt area and its impacts as a
function of global temperatures (Burton, Lampe et al., 2024) is currently being explored as a future development. This could
form part of a broader "biodiversity nexus", addressing a sparsely represented arena in integrated assessment models; this
could include effects such as changes in land use areas under climate change, direct impacts on livestock, and effects of
ocean temperature and acidification changes on aquaculture. In this way, FRIDA could represent key global drivers of
biodiversity loss. Another nexus, relating to human health, could seek to extend the current representation of climate impacts
on mortality to broader effects on morbidity, with potential extensions to wider conceptions of wellbeing. Other currently
excluded impact channels include effects on and consequences of water scarcity, flood damages, and extensions of Sea Level
Rise damages to wetlands and ports (see Ramme et al. (submitted)).

Additionally, the deterministic climate response in FRIDAv2.1 could be developed to include a stochastic component, based
on the representation in the FaIR model on which the Climate module of FRIDA is substantially based (Smith et al., 2024).
Representation of threshold-based nonlinear climate responses such as melting permafrost, which may generate substantial
carbon dioxide and methane emissions, could be incorporated in future development. This concept could be extended to
internally model "tipping points" in both the Earth and social systems - which could be explored in FRIDAv2.1 in bespoke
scenarios, but are not internally modelled due to the uncertainty of their thresholds (McKay et al., 2022) - while taking care
to reflect uncertainties and cascading effects appropriately.



The third category contains climate impact channels identified by the IPCC which it does not seem possible to incorporate in
a global-scale aggregated modelling framework such as FRIDA, whether due to their high and unquantified uncertainty, their
strongly regional dynamics, their dependence on detailed processes, or their relatively small global effect relative to their
complexity. For example, any future "biodiversity nexus" as discussed above, if implemented at the global scale, would only
enable a high-level representation of some key aspects of this system component, in particular global drivers of biodiversity
loss rather than metrics for biodiversity itself. Highly regional channels such as pollution, habitat fragmentation, tree
mortality, pest outbreaks, weeds, disease occurrence, species loss, and effects on marine ecosystems, as well as food-systems
effect such as food diversity, nutritional quality, access, and safety, and pollination changes from species loss would likely
be excluded. It is worth noting that these highly specific and process-detailed biodiversity and food-related impact channels
are generally not currently represented in the state-of-the-art land systems models. This currently negates the possibility for
inclusion via damage functions trained on land systems models, and contextualises their exclusion at the process-level in
FRIDA.

Similarly, a "human-health nexus" would only incorporate idealised representations of this complex, highly regionally-
disaggregated system component. While aggregated effects on water scarcity may be implemented under future
development, effects of climate changes on water quality, water-borne diseases, and local flooding will likely not be
addressed in a model of this nature.

Finally, while climate change is expected to affect the dynamics of migration and conflict, these effects are highly uncertain
and act to modulate other key complex, unmodelled drivers of these processes. It would therefore be infeasible to include
these complex, highly regional dynamics within the global framework of the FRIDA model.

It should be noted that the scope of these categories is dependent on the model structure, and so any future development of
the broader structure of FRIDA will alter this allocation. For instance, if the model is regionalised, more spatially-
disaggregated impacts, or impacts which directly affect the connections between regions, may be plausibly represented in the
model. Additionally, the level of detail in damage functions derived from other models is dependent on those models'
representation of the relevant processes; utilising data from updated models featuring additional processes will facilitate the
incorporation of these effects, at the abstracted level, within the FRIDA framework.

## 3.2 The methodology for representing climate damages in FRIDAv2.1

The FRIDAv2.1 model operates at the global scale, with no regionality. While this is justified on the basis of FRIDA's focus
on broader human-Earth system-wide trends rather than detailed local analysis, it should be noted that this global-scale
treatment necessitates the neglecting of important spatial inequalities of climate change and associated impacts (Burke et al.,
2015; Méjean et al., 2024; Palagi et al., 2022; Waidelich et al., 2024). It should also be noted that aggregated modelling may
allow for the partial cancellation of uncertainties in individual climate impacts, as Müller et al., (2021) found for crop yield
responses. In addition, representations of global damages may be useful for aggregated analyses of future scenarios in cost-
benefit or similar analyses.





Bottom-up modelling of climate impacts benefits from transparency and legibility, with clear attribution of climate damages across impact channels. However, the piecemeal nature of this approach risks neglecting impact channels, or double-counting impacts. As documented in Section 2, care has been taken within FRIDAv2.1 to avoid double-counting, through the use of conceptually consistent literature choices. As discussed in Section 3.1, while FRIDAv2.1 contains many key impact channels, many are excluded, and while some may be incorporated in future development, several will likely not be contained in the FRIDA modelling framework.

Several important climate impact mechanisms remain unrepresented in the FRIDA model, in keeping with concerns around bottom-up climate damage representations. However, the significant number of important channels systematically represented within the model provides a substantial step in incorporating the coupled effects of climate damages in such an IAM. It should be noted that, since several impact channels are included in the internal model calibration, a component of impact mechanisms not explicitly represented should be present via these channels, such as the failure rate of loans, to the extent that these impacts are present in the historical calibration data.

Where possible, climate impact channels were incorporated based on existing literature, with parameters taken from fits to existing data, whether empirical or modelling-based. However, for several important channels this was not possible, necessitating these parameter values to be set via optimisation within the internal calibration of FRIDA. These impact channels are therefore candidates for study in more detailed and specific models, which may reveal them to be the combination of multiple more specific and therefore more readily measurable impacts. Future development of FRIDA will seek to replace these as literature allows; the inclusion of other literature-based impact channels can also be expected to reduce these calibrated channels' effects during calibration.

Reviewing the list of climate damages compiled in the Technical Summary to the IPCC WGII AR6 ensured a systematic methodology in the approach. This allowed for the consideration of as many climate impacts as possible; while many of these were precluded from inclusion in FRIDA due to the model's scope (see Section 3.1), this represents a consistent approach to a strategy for representing climate impacts in the model. It should be noted that despite the vast array of literature considered by the IPCC WGII, their list of impacts may not be exhaustive, with missing impact channels therefore not considered for inclusion here.

The separation between future scenarios and the effect of climate impacts in the ScenarioMIP framework, including in CMIP7 (van Vuuren et al., 2025), while justified on methodological and logistical grounds, renders the coupled effect of climate impacts unexplored within ScenarioMIP, and therefore allows for the creation of scenarios which may be highly unlikely; for example the overall dynamics of SSP1, SSP2 and SSP5 baselines are difficult to recreate in FRIDAv2.1 (Schoenberg et al., submitted.). Initiatives such as FRIDAv2.1, with its representation of climate damages, can therefore provide important insights into the future development of the human-Earth system.

To couple climate impacts into an IAM such as FRIDA, links must be created between outputs of the Climate module and variables within the other model components. Since FRIDAv2.1 is a global model, the drivers of climate impacts are global mean quantities, namely temperature, sea level rise, and $CO_2$ concentrations. There is currently no stochastic internal



variability within the climate system in FRIDA, though this will be explored in future development (Section 3.1). For a given set of climate parameters, the same deterministic climate trajectory follows from the same set of emissions. The consideration of internal variability has been found to drive differing impacts when using nonlinear damage functions (Calel et al., 2020; Schwarzwald et al., 2022), though this will be dampened on the global scale.

While climate impacts in FRIDA are necessarily represented as functions of aggregate, globally averaged climate variables, these effects are in reality local, and mediated by specific physical events such as storms and heatwaves. Where possible, this paper describes these associated local drivers of given impact channels, which are not directly modelled in FRIDA but implicitly represented through their association with global mean values. Local impacts and extremes remain important, and will be considered should FRIDA be regionalized in a future version.

While the variety of SLR-driven impact streams is unprecedented in a globally aggregated IAM, there are also a range of remaining SLR impacts that are not accounted for. Coastal erosion will increase under future SLR (Cazenave and Cozannet, 2014). The construction of flood protection reduces wetland areas, and the land that is used for flood protection or abandoned potentially has an opportunity cost (Wong et al., 2022). Furthermore, the coastal zone could have socio-economic dynamics that are different from the global average, for example leading to relatively higher growth in coastal population or

asset stocks, which could increase future damages even further. These other impact streams have not been included in FRIDAv2.1, because they do not necessarily have a corresponding impact channel in other parts of the model, or they are difficult to model in a globally aggregated setup.

No "tipping points" in the coupled human-Earth system, or in specific impact channels, are explicitly represented in FRIDAv2.1; the functions presented in Fig. 1 and Table 1 are smooth and continuous. This is due to the large uncertainties in

the specifics of any threshold temperatures above which tipping points may occur (Section 3.1; McKay et al., 2022). A candidate representation of Marine Ice Cliff Instability tipping has been implemented, but remains switched off in the default mode due to these uncertainties. The flexible nature and short runtime of FRIDA means that implementation of threshold-based tipping elements would be straightforward, either if more robust evidence of these thresholds emerges, or simply as a sensitivity experiment, informed by existing literature estimates such as TipMip (https://tipmip.pik-potsdam.de/), McKay et

al., (2022), and Wunderling et al., (2023). Furthermore, future development work within the project will investigate tipping-like behaviour in two Earth System models run to 2300 under overshoot and reversibility scenarios, which may inform any future representation of tipping points in the model. It should be noted that non-linear interactions between components of the system remain a key part of the FRIDA model (Schoenberg et al., submitted.), allowing for the emergence of tipping elements within the coupled evolution of the model, even without the explicit inclusion of threshold-based physical climate

impact tipping points.

While the inter-module feedbacks due to climate change are documented here, some other connections from climate variables are present in FRIDAv2.1, noted here for completeness. Firstly, internal climate feedbacks are represented in the model. These occur in the carbon cycle (effects of temperature and $CO_2$ on NPP in grass and forest, and of temperature on the lifetime of soil carbon and the growth of forest biomass, as well as effects on ocean $CO_2$ uptake) and the atmospheric



lifetimes of various radiative species; these are described as part of the climate module as documented in Wells et al. (in prep). Secondly, outputs from the climate module are used to provide indications of changes in the biosphere, conceptualised as transgressions of the widely utilised "planetary boundaries" (Richardson et al., 2023). As described by Eriksson (2025), these use the atmospheric $CO_2$ concentrations and anthropogenic effective radiative forcing as indicators, but do not feed back to the rest of the system, and therefore are not characterised as climate impact channels here.

The approach to uncertainties in the climate impact parameters varies between the impacts which are literature-based and those which are internally calibrated (see Table 1). Where uncertainty is extractable from the literature for a channel, the uncertainty is sampled to give 11 parameter sets equally spaced between the 2.5th and 97.5th percentiles. For the internally calibrated sets, the uncertainty is derived from the calibration. Upon simulating the full ensemble - varying each model parameter across its uncertainty range - the ensemble is constrained to approximately reproduce historical observations,

removing the most unrealistic samples (Schoenberg et al., submitted.). This approach is not fully consistent across the impact channels, with qualitatively different approaches to different impact processes, reflecting the limitations within the existing literature. This serves as a call to the wider climate impact community for the generation of a broader set of internally consistent, readily applicable damage functions which can be utilised in IAMs. In addition, since some impact channels are unlikely to be implementable in a model like FRIDA, specific research into the effects of those channels is also crucial.

Relatedly, the damage functions generated as part of the process of implementing these impact channels in FRIDA, both literature-derived and internally calibrated, are general enough so as to be utilised in other modelling frameworks where appropriate.

### 3.3 Adaptation

Reductions in the level of climate impacts for a given level of climate change, termed adaptations, can play an important role

in the dynamics of the human-Earth system trajectory (e.g. Colelli et al., 2022), with substantial uncertainty (e.g. Molina Bacca et al., 2023). Adaptation measures are historically underrepresented in IAMs, due to their high heterogeneity and dependence upon prior representation of climate impacts (Andrijevic et al., 2023; Holman et al., 2019; van Maanen et al., 2023). Currently, due to the initial focus on representing climate impact channels, few active adaptation mechanisms are represented in FRIDAv2.1.

Due to the high level of abstraction in FRIDA, some forms of adaptation to particular climate impact channels may never be explicitly represented, similarly to several climate impacts themselves. This aspect of modelling has not been fully explored in FRIDAv2.1, and will be extended further in updated versions, potentially using aggregated concepts such as adaptive capacity (Andrijevic et al., 2023). In addition, some externally-imposed climate adaptation measures can be explored as part of a scenario analysis. Finally, the uncertainty in the magnitude of the climate impacts can also be utilised to study the effect

of weaker climate damages than the mean response, as a proxy for adaptation strategies.

The level of representation of explicit adaptation measures is generally described in each subsection in Section 2; proactive adaptation is either explicitly represented (sea level rise; Section 2.5.4 and Ramme et al., (submitted)), implicitly partially



included under other drivers of the downstream variables (crop yields, risk perception and adaptive behavioural responses), can be explored using externally imposed drivers (e.g. concrete), has plausible routes to relatively simple representation in

future versions (labour productivity, energy supply), or remains unaddressed within FRIDA (e.g. temperature-driven mortality).

It should be noted that, due to the highly coupled nature of FRIDAv2.1, the occurrence of climate impacts drives cascading responses throughout the system, in ways which are often framed as forms of autonomous or implicit climate impact adaptation (Colelli et al., 2022; Holman et al., 2019; van Maanen et al., 2023). For instance, a balance between supply and

demand for energy, food, and water is enforced; climate-linked changes in these (from e.g. impacts on energy supply, energy demand, crop yield, and evapotranspiration) is met via compensating changes in those systems, which have associated financial and other costs. Similarly, changes in economic damages - e.g. the indirect economic effects on lending standards (Section 2.5.2), or impacts on labour productivity (Section 2.5.1) - drive changes through the tightly-coupled economic module, affecting investment decisions in ways which represent a form of adaptation. Infrastructure damage filters through

in a similar way, with rebuilding of damaged structures as a response to damage, without a proactive adaptation measure.

## 4 Conclusions

This paper described the implementation of 16 climate impact channels across 12 categories in FRIDAv2.1 (Fig. 1). At least one impact channel has been linked to each module within FRIDA - Land Use & Agriculture, Economy, Demographics, Energy, Behavioural Change, and Resources, with varying levels of process detail. Where possible, the parameters for these

impacts are constructed using and condensing estimates from the existing literature; in other cases, the parameters have been calibrated within FRIDA, ensuring the model reproduces historical (1980-2023) behaviour.

These changes will allow the exploration of many broad questions within FRIDA, focusing on the system-wide effect and interactions of one, several, or all of the impact channels documented here. The process-based framework followed in this model allows for more detailed understanding of their effects than the simplified representations in many IAMs, including

nonlinear hysteresis effects (Diaz and Moore, 2017).

While many key climate impact channels have been represented in FRIDA (see Section 2), there exists a vast array of potential climate impacts (Pörtner et al., 2022), many of which remain unexplored in the model. These have been systematically analysed, and filtered into a set of climate impacts which will not be represented in a global, aggregate model such as FRIDA, and a set which could plausibly be included and which will therefore be investigated (Section 3.1). Future

work will assess the dynamics of the impact channels currently represented, and also explore the inclusion of these additional effects. In addition, specific experiments utilising these damage functions to determine the potential response to climate forcers such as geoengineering and large volcanic eruptions will be investigated.

The process undertaken to represent climate damages within FRIDAv2.1 reflects a shift in the philosophy for simulating future scenarios in a fully coupled IAM, with climate impact channels - connections between the climate and other human-



Earth system components - set on the same conceptual level as interactions between and within other parts of the system. The damage functions condensed as part of this process may be utilised in other modelling frameworks, working towards a clearer understanding of the dynamics of the coupled human-Earth system.

**Code and Data Availability**

The literature-derived parameter sets and code used to process Figs. 1 and A.1 can be found in Wells (2025). The
FRIDAv2.1 model and additional information can be found in Schoenberg et al., (2025).

**Author Contribution**

Conceptualisation: all authors. Funding acquisition: WAS, CS, CM. Investigation: CW, BB, LR, JB, BC, AM, JKR, ANL, AEE, WAS, and LWE. Methodology: all authors. Writing - original draft: CW. Writing and editing: all authors.

**Competing Interests**

The authors declare that they have no conflict of interest.

**Financial Support**

This research was supported by the Horizon Europe research and innovation programs under grant agreement no. 101081661 (WorldTrans).



# Appendix A

## A.1 Climate Damage Functions



**Figure A1: Panel plot of the climate impact channel functions displayed in Fig. 1 and described in Sections 2.1-2.6. Red curves represent the median central estimate, with grey giving the uncertainty variation. In cases in which uncertainty is treated by**



**"Sampled Percentiles" in Table 1, the 11 parameter sets representing the 2.5th to 97.5th percentiles, with equal spacing, are shown; for the other impact channels, the range using the upper and lower parameter values after literature analysis and internal calibration is shown.**

## A.2 Functional Forms

Equations A1-A3 show the functional forms applied to multiple damage functions, and referred to in Table 1. $D$ represents 910 the damage response; $T$ represents global mean temperature anomaly; and $C$ represents global $CO_2$ concentrations. A1 and A2 have zero intercept by design, since the damages at pre-industrial temperatures were conceptually assumed to be zero.

$$D = a\Delta T + b(\Delta T)^2 \tag{A1}$$

$$D = a\Delta T \tag{A2}$$

$$D = a + b\Delta C + c\Delta T + d(\Delta T)^2 \tag{A3}$$

## A.3 Energy Supply

The temperatures used for generating the impact functions for thermo- and hydro-electric power were derived from the climate data used to drive the impact model, a process made more complicated due to the specific climate data used. The impact models were driven by RCP2.6 and RCP8.5, using ISIMIP FastTrack data. This data was produced using five GCMs, 920 and the multi-model mean data, along with the minimum and maximum, was reported by Van Vliet et al., (2016). However, the climate data was only produced from 1961, meaning that temperatures relative to pre-industrial levels are not available. To account for this, the later ISIMIP2b historical data, available since 1861, was used for each model to estimate the offset between approximate pre-industrial and recent history (i.e. 1961 to 2000 minus 1861 to 1900). This offset was applied to the FastTrack data, with the 1961 to 2000 average subtracted, to give an approximation of the trajectory that would have been 925 attained relative to pre-industrial times. Of the five GCMs used in the FastTrack data, three (IPSL-CM5A-LR, HadGEM2-ES, and GFDL-ESM2M) were also included in ISIMIP2b, making the combination of this data appropriate. One model had different versions between the simulation rounds (MIROC-ESM-CHEM in FastTrack, MIROC5 in ISIMIP2b), which was assumed consistent for this purpose, and one (NorESM1) had no counterpart in ISIMIP2b. For NorESM1, the mean across the other four models was used. This method is therefore an approximation of the actual temperature relative to pre-930 industrial levels, but any negative effects of this approximation are dampened by the continuity of most of the models between the rounds, and the small level of climate change from pre-industrial times to the mid-20th century.

With the global temperatures offset to pre-industrial levels and their associated climate impacts on thermo- and hydro-electric power generation obtained, a quadratic fit with no intercept was performed to emulate the impacts relative to pre-



industrial levels (Equation A1). This functional form was applied separately to power generation from thermo- and hydro-
electric plants.

The data was provided for the mean, minimum, and maximum from five models. Since all climate impact uncertainties
should be expressed on a common range - in FRIDA, taken to be the 2.5-97.5 percentile range - and in the absence of
additional uncertainty information, it was assumed that the five models uniformly sample the uncertainty space, with the
minimum and maximum response representing the 17th and 83rd percentiles respectively. Assuming a skew-normal
distribution at each temperature, the corresponding quadratic fits for the 2.5 and 97.5 percentiles were determined. The
effects of climate change on thermoelectric plant efficiency via these effects is negative, with impacts from 3-11% at 2°C of
global warming.

### A.4 Energy Demand

Data from five Earth System Models covering three future scenarios (SSP1-2.6, SSP3-7.0, and SSP5-8.5) were utilised as
per Werning et al., (2024b, a). Modifications were made to the procedure of Werning, Frank, et al., (2024) to generate
population-weighted values of CDD and HDD.

Figure A2 shows four fits to the simulated global population-weighted CDD and HDD - linear, quadratic, exponential, and
logarithmic - along with their residuals.

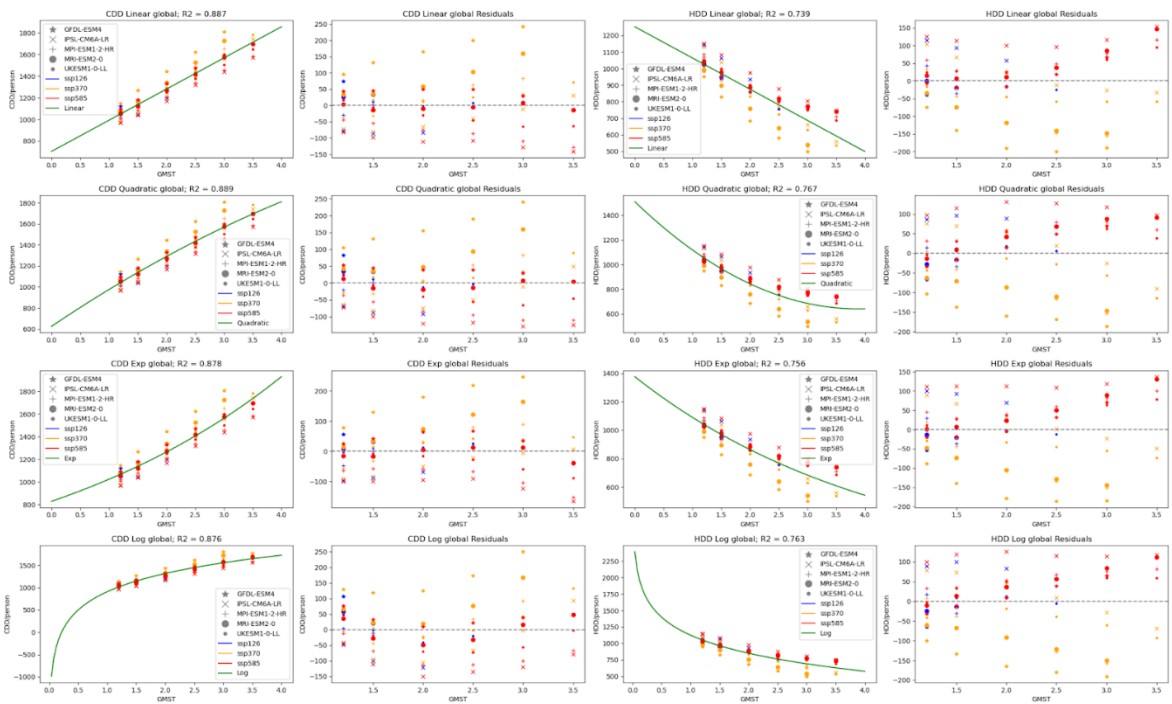


**Figure A2. CDD and HDD expressed as functions of GMST, with different functional fits shown along with their residuals.**



Based on these fits, for use in FRIDA a linear function in GMST was applied for population-weighted CDD, with an exponential function for population-weighted HDD:

$$CDD = a + b\Delta T, \tag{A4}$$


$$HDD = ae^{bT}. \tag{A5}$$

### A.5 Exposure to Climate Extremes

The population-average exposures are shown against GMST in Fig. A3. Variation between scenarios is due to systematic differences in population distribution, as well as internal variability. Note that, uniquely, the Warm Nights metric deviates
between scenarios from the datapoint corresponding to the 2020s, when the population and climates are near-identical. The cause of this is unknown and so this metric's data was excluded from the analysis. The seven remaining metrics utilised are then (see B. Li et al., (2023) for exact definitions) PRCPTOT (Annual total precipitation), RX1D (Maximum 1-day precipitation), R50 (Number of days with heavy precipitation), WD (Warm days), HW (Heatwave), SPH (Sequential Precipitation-humid heatwave), and CDHW (Compound drought and heatwave).

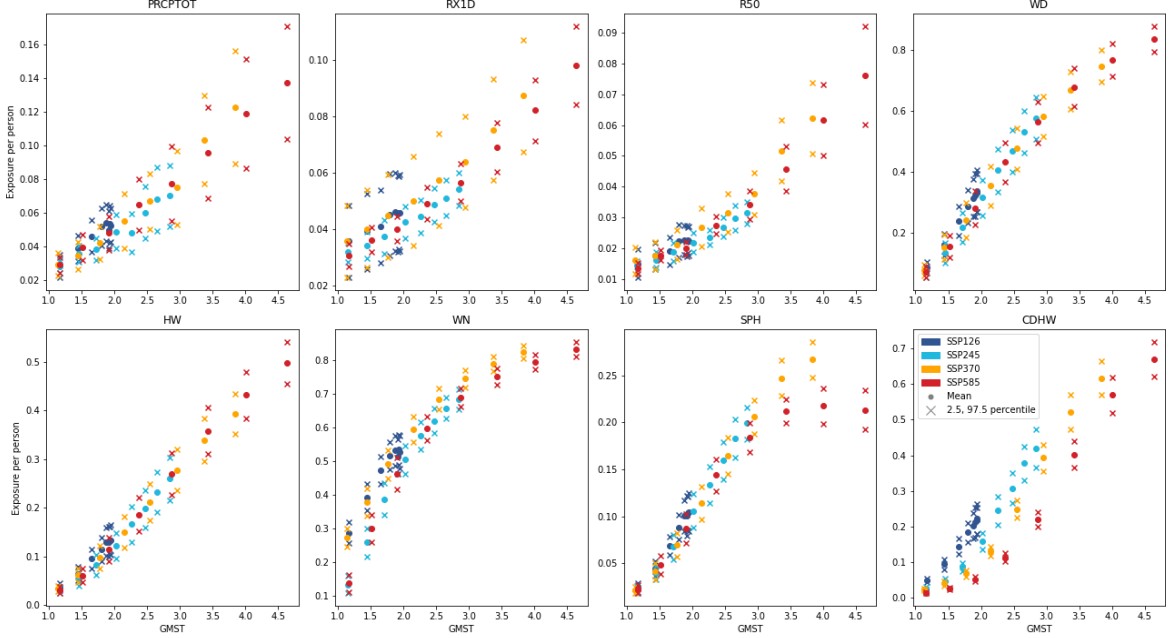


**Figure A3. Changes in population-weighted exposure to extremes of various climate metrics, under multiple scenarios, as a function of global mean temperature. The mean and 2.5th and 97.5th percentile values across 21 climate models are shown.**





The functional form of Equation A1 is used for the fit between the exposure metric (the average number of indices which a person experiences at least one event of in a given year which is more extreme than the historical record) and global mean temperature.

## A.6 Mortality

The functional form of Equation A1 is used for mortality impacts, with the damage D representing the fractional change in mortality. Accounting for the age-dependence as per Chen et al., (2024), using 2024 population levels, FRIDA scales the temperature effects by the following factors for cold and hot mortality responses: 0.9169 and 0.9651 (for 0-65 years); 1.742 and 1.1581 (for 65-75 years); 2.2004 and 1.7372 (for 75 plus).

## A.7 Labour Productivity

Data for the impact of temperature on labour productivity was traced from Dasgupta et al., (2021)'s Fig. 5 and Supplementary Fig. 5 for Low and High exposure respectively. Then Equation A1 was used to project the resultant relative percentage change in labour productivity.

While Dasgupta et al., (2021) present uncertainty ranges, it seems these represent variation in grid cells within regions - with the uncertainty bars for the global mean representing the variation spatially across the globe - and are therefore inappropriate for use in the global FRIDA model. Instead, the median values were used in FRIDA. In lieu of a reasonable estimate of this uncertainty, but with the knowledge that such an uncertainty exists and may be substantial, uncertainty in these parameters is explored as part of the broader model calibration, which is detailed in Schoenberg et al., (submitted).

Dasgupta et al., (2021)'s results were generated by categorising sectors of work in the survey data into the Low and High exposure categories; for FRIDA, then, this same sectoral mapping should be applied. Raw labour supply in FRIDA, measured as the number of people in employment, was first distributed between the three sectors agriculture, industry, and services, a common structural split (Leimbach et al., 2023; World Bank 2012) and the three sectors modelled in FRIDA. The fraction of employment in each sector was modelled as a function of GDP per capita, using historical data from the World Bank (2012). Generally, increasing GDP per capita drives a reduction in agriculture and an increase in services, with industry's share peaking and declining. Then, these sectors were mapped to the exposure types from Dasgupta: Low, High, and the zero-effect No exposure. The fraction of each sector represented as No/Low/High exposure was 0/0.1/0.9 for agriculture, 0.25/0.5/0.25 for industry, and 0.75/0.25/0 for services.

## A.8 Crop Production

Changes in total crop production were represented using Equation A3. The absolute temperature (in Kelvin) is used for $T$, converted from temperature anomaly within the Climate module, using the 1850-1900 value to account for the offset. The 1850-1900 absolute temperature is estimated by subtracting the 1850-1900 to 1961-1990 difference in the IPCC AR6 global



mean surface temperature (GMST) from the HadCRUT absolute temperature estimate for 1961-1990 (Osborn et al., 2021). Absolute values of $CO_2$ concentration are also utilised.

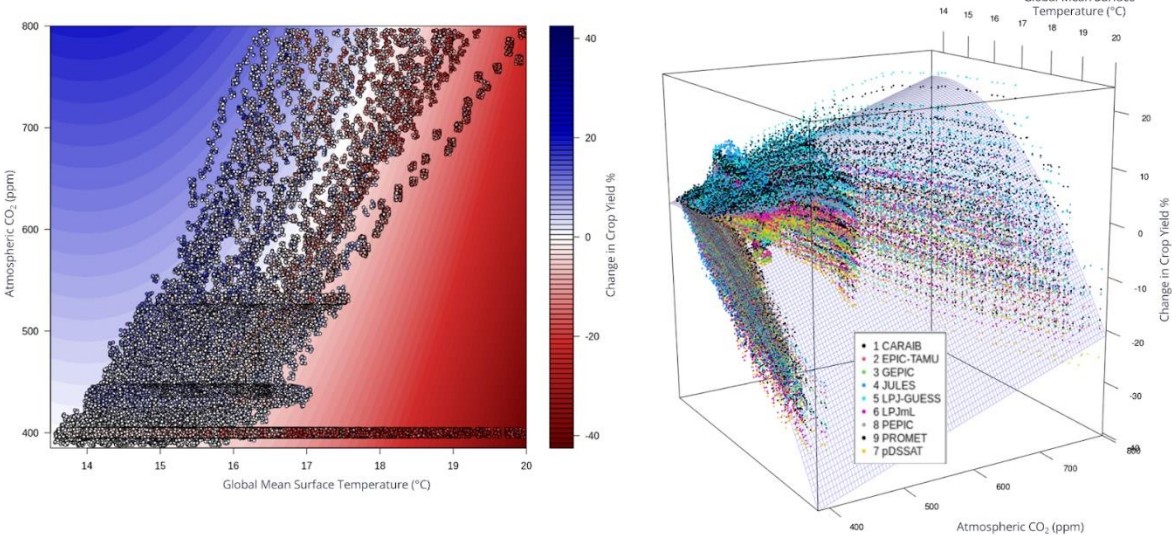

**Figure A4. Response in crop yield as a function of global mean temperature and $CO_2$ concentrations. Individual data points represent climate model-crop model combinations from** Müller et al., (2021)**, annually resolved. Data from multiple future SSP scenarios as well as idealised fixed-$CO_2$ versions are shown. Left panel shows the $CO_2$ - GMST combinations, with the chosen fit shown in coloured contours, and the annual model data as coloured datapoints. Right panel shows the results in 3D space, with the yield response in the 3rd dimension.**

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
