# Peer review of "The Representation of Climate Impacts in the FRIDAv2.1 Integrated Assessment Model"

_EGUsphere, 2025_

## Author Comment (AC1)

We thank the reviewer for taking the time to explore the manuscript, and for their points raised. While overall we disagree with the reviewer's framing of the manuscript and the wider literature, we appreciate the engagement and hope we have provided suitable responses to each point - including additional text to the manuscript - which we address in turn below.

"Estimates of the total impact of climate change often attempt to measure or model the overall impact on global Gross Domestic Product." The typical impact is a Hicksian Equivalent Variation, rather than an income or output effect.

We appreciate the reference to the broader context, but note that we are here referring to existing literature, and therefore prefer to frame this in the terms used therein, which overwhelmingly utilises GDP effects. We use a process-based approach to climate damages, rather than a top-down GDP one, and therefore consider it unwarranted to explore this noted distinction in this manuscript.

You can't cite Burke without also citing the by now many papers that make mince meat of that paper.

While we understand the critiques of the top-down economic damages approaches used by e.g. Burke et al., 2015 - and indeed adopt a bottom-up process-based representation of climate impacts partly on this basis - our assessment of the overall literature is not the one presented here. We didn't see it necessary to present in further detail the pros and cons of a methodology which we haven't used in our modelling approach.

The set of included impacts is haphazard. Most impacts are based on a single study, ignoring most of the literature. Some impacts are based on primary research, of hair-raising quality.

In the manuscript, we set out in detail the process by which we arrived on the set of climate impact channels modelled in FRIDAv2.1. As part of this discussion (Sections 3.1 and 3.2), we fully acknowledge that this set is an initial formulation of the damages under climate change, constrained by existing literature and data. However, and especially in the context of existing modelling frameworks which exclude climate impacts by design, we must insist that the goal should be the incorporation of as wide a set of climate impact channels as reasonably possible, in order to better simulate the fully coupled system.

The sections on indirect economic effects and government spending would make an undergraduate blush.

The previous review comments, while abrupt, focus on points of substance within the manuscript, and we have therefore responded to their substance. We feel however that this comment doesn't engage with any specific substantive issue with the manuscript, and as such we are unable to comment or suggest modifications to the paper on this basis.

All impacts ignore Schelling (1984). Why on Earth would impacts be a function of climate change and climate change only? What happened to human agency, technological progress, institutional change?

We fully agree that the effects of climate impacts are not purely a function of physical drivers, with the overall effect dependent on the underlying context. Indeed, part of the motivation for using a coupled System Dynamics model like FRIDA is that variations in the response to feedbacks can occur based on the properties of other system components. Within FRIDA, this occurs in several domains - via e.g. costs of adaptation, variation in the

behavioural response, and energy source diversity. In addition, agency and institutional change are best incorporated via external scenarios.

However, it is a definite limitation of the modelling framework, constrained as it is by the existing climate impacts literature, that the effects of socioeconomic factors are underrepresented. We feel strongly that, due to the uncertainties at hand, this context-dependent nature of climate impacts should for now be explored as part of the uncertainty ensemble, coupled with external scenario imposition, in bespoke experiments. On this basis, we have added the following text to the manuscript at L790:

*It should be noted that the climate impact functions in FRIDAv2.1 are driven by purely climate variables, whereas their overall effect can be expected to depend on the broader socioeconomic context, such as the level and form of technological development, and shifts in institutions and societal perspectives. Within the current FRIDA framework, these effects are best explored downstream of the damage functions, with some aspects of socioeconomic effects (e.g. technological change) suitable for analysis within the parameter uncertainty framework (see Schoenberg et al., 2025), and other, more qualitative aspects (e.g. institutional and decision-making changes) best studied using exogenous policy impositions. Future development of FRIDA can seek, where possible, to endogenise these effects into the internal damage functions.*

We appreciate the highlighting of this oversight in the original manuscript, and feel the paper is better orientated as a result.

---

## Author Comment (AC2)

We greatly appreciate the time and effort put into these detailed reviews; specific responses are detailed below in blue text. Line numbers refer to the document with tracked changes; in many cases we quote directly from the new text (in italics) for clarity.

**Reviewer 2**

The authors have taken on a difficult task, to present a set of climate impacts linked to an existing integrated assessment model (IAM) called Feedback-based knowledge Repository of IntegrateD Assessments version 2.1 (FRIDA v2.1). The purpose aligns with contemporary scientific needs and research as there is currently substantial effort, by many, to quantify climate impacts and feedbacks between human-earth, human-environment, and human-ecological systems at a variety of scales (some of which coincide with the different h-e terms). In many cases, entire manuscripts focus on only one impact and here the authors try to weave 16 climate impacts through 6 IAM modules for which the reader may have little to know knowledge of FRIDA. The authors have set themselves a nearly impossible task, for a single manuscript, which is to demonstrate scientific contribution and novelty/significance, rigor/quality, and ensuring comprehension by the reader and reproducibility by those highly interested in the content.

Based on the criteria set forth by the journal and their use of the geoscientific model development review criteria, I do not believe the manuscript in its present form is ready for publication and major revisions or a revise and resubmit is required. Below I outline my major concern followed by some minor issues.

Many thanks for this detailed feedback on our manuscript; we feel the incorporation of these points has greatly improved the manuscript in content and clarity.

With reference to the general point, we acknowledge the difficulties inherent in this documentation process, but we also consider it essential to document the impact channels that we incorporate in this version of the FRIDA IAM. We grappled with how best to do this, and had considered splitting this into separate submissions, but we felt that combining the overall set of impacts in Figure 1 allows the reader to get a clearer overall sense of the set of impacts implemented, with the discussion also facilitating an overview consideration of the process of their implementation – the documentation of which we consider almost as important as the documentation of the specific impacts. In addition, we felt that there was not a clear clean split to balance the number of papers with their detail and overall coherency.

We feel we should also note for clarity that here we are not incorporating climate impacts into an existing IAM; the philosophy of the model is such that these impact channels are part of the set of overall systems feedbacks used to drive the coupled human-Earth system. This is explored further in the overview paper Schoenberg et al. (referred to as submitted in the original submitted manuscript, but this has now been accepted for GMD), and other papers within this GMD collection (please see below for clarification on this collection). As such, this paper represents part of the model documentation by which users can understand the component parts of the model. We have clarified this point now in Section 1 (see below).

We have detailed our changes based on these review comments below, including substantial reorganisation of parts of the manuscript to aid the paper flow.

**Major concerns**

Flow and scope. The manuscript should stand alone without having to read other documents and I did not feel like that was the case. It was also a very slow read with a weak narrative, which led to further loss of enthusiasm. It might be more interesting to reduce the presented set of impacts, highlight their contributions, and provide results that demonstrate their credibility. Consider the potential impact beyond FRIDA.

We accept and agree that the slow flow of the draft manuscript limited the ability of readers to fully engage with the overall content. This is a challenge in any detailed model documentation, which attempts to describe a coherent subset of the overall model without losing the documentation of connections to the rest of the model, especially with this paper as it documents processes which by definition connect across multiple modules of the FRIDA IAM.

In terms of the set of impacts presented, as discussed above, we feel that all climate impacts in FRIDA, including their process of selection and simulation, should be documented together. To address these concerns, therefore, we have made substantial modifications to aid the flow and ensure the manuscript more coherently stands alone. Many of these are informed by the below suggestions on specific and general changes to improve the flow of the text (see the responses to those points for specifics). We have substantially revised the introduction and discussion sections to better set out our approach and situate it in its wider context.

Specifically in terms of the paper scope, and to incorporate this into the paper flow, we have substantially modified the last paragraph of Section 1. Firstly, we set out the paper's scope, and secondly indicate that the start of Section 2 gives an overview, with Sections 2.1-2.6 giving the full detail in each module – this will allow readers to more easily overview the impacts, and then to pick specific impacts of interest. The full paragraph (from L163) now reads:

The purpose of this paper is to document both the representation of climate impact channels in FRIDAv2.1, and the wider methodology for their selection and construction. Included in this is discussion of the represented set of impact channels in the context of FRIDA development. Analysis of future scenarios in FRIDA and the role of climate impacts in the evolution of the coupled system remains out of scope here, due to the detail needed in that analysis and the need to fully document all climate impact channels here; this topic will instead be explored in future studies on FRIDAv2.1. The paper is structured as follows: Section 2 firstly introduces the full set of climate impacts in FRIDA, with Sections 2.1-2.6 then describing the specific implementation of the climate impacts represented in FRIDAv2.1 at the module-level, with additional detail provided in the appendix where necessary. In general, for each identified impact channel, relevant literature was explored to identify possible routes for inclusion into FRIDA. Where possible, uncertainties are given for the 95 percent confidence (2.5th-97.5th percentile) range. The methodology and the broader context - including the possibility of utilising these impact functions outside the FRIDA IAM - are discussed in Section 3, and Section 4 provides conclusions.

We have made several alterations which aid the paper's narrative strength, such as a stronger framing in the introduction (L54):

To operationalize this, FRIDA is designed as a system dynamics-based coupled human and natural systems (CHANS) model building on prior system dynamics approaches to human-Earth modelling (Schoenberg et al, 2025).,

a further motivation and situation with respect to other IAMs (L117; "Other IAM frameworks have adopted...")

and especially later in the manuscript, the new Section 3.3 Contributions of FRIDA's methodology for representing climate damages (see below).

Impact Contributions. The way the text was written it was very difficult to distill any contributions from the representation of the 16 impacts. Each impact section (e.g., Section 2.2.1) it felt like I was sifting through content (in this example four paragraphs) to find out what was done in the second last paragraph and how the impact was related to sea level rise in the last paragraph. While the front matter could have been useful, it didn't motivate the approach used by the authors AND there was insufficient information and details about each impact approach to provide the reader with confidence or to enable replicability of the work. It would be more interesting and impactful to showcase the contribution of the impact approach first, provide information about how the impact was quantified so that others could do so similarly, and then provide additional supporting information if required.

We wholeheartedly agree on the need to balance the level of detail and to optimise the order of information presentation to engage the reader and provide them with the most useful information. We have therefore reorganised some of the subsections of section 2 in order to more clearly present the motivation and methods used.

In particular, we now ensure that a short summary of the approach in FRIDAv2.1 is given in the first paragraph of each subsection in order to give the reader the key information at the earliest opportunity, before discussing the methodology further (e.g. Section 2.1.1: "Based on existing literature estimates, in FRIDAv2.1 only climate impacts on thermoelectric and hydroelectric power generation are simulated."; Section 2.5.1 "This impact is included in FRIDAv2.1 using Dasgupta et al., (2021) to simulate reductions in productivity by sector under global warming.").

We have reorganised and simplified the sections energy system and behavioural impacts of climate change. For example, in *Section 2.1.1 Energy Supply* we have focused firstly on the changes we implemented (on thermo- and hydro-electric power) before discussing wind and solar power (which see no damages in FRIDAv2.1); in *Section 2.1.2 Energy Demand* we have condensed the section on HDD and CDD metrics, removing surplus parts on methods to calculate these which were not utilised in FRIDA; and in *Section 2.2.1. Exposure to Climate Extremes* we removed unnecessary detail on historical changes in datasets which we do not use here. In doing so, we have endeavoured to focus on the motivation for the specific methodology chosen here.

Finally, we are uncertain on the specific meaning of your comment on "the contribution of the impact approach"; we interpret it as suggesting that we explore the relative importance of each climate impact in Section 2 (a point also made by Reviewer 3). If this is the case, we note that we have added the new figure A2 in response to one of your comments below, which now shows the reader the response of each climate impact channel under a baseline emissions scenario. In addition, as noted above we have implemented a new statement describing the paper's scope (and that future scenario analysis is out of scope) beginning on L163, noting that we feel it would be unsuitable to present future scenario analysis here on the basis of the scope of this documentation paper and the complexity of the model response.

Paper Contributions. The authors almost immediately reference FRIDA in the introduction and the way the document is written limits the readers interest because it is written with a FRIDA

focus. There are no research questions and little discussion of other IAM impact or non-IAM impact representations. It would be more informative to the reader and provide more credibility to the impact approaches to present the impacts in a way that might inspire other modellers to represent your presented impacts in their research. Sure, because you've integrated your impacts with FRIDA you have some constraints, but I'm sure it would be possible to suggest different types of data for inputs. As it stands, it seems like the paper is written for FRIDA users only.

We had noted that these functions aggregated here may well be useful to others (e.g. towards the end of the abstract), but we agree that we should be more explicit and provide more discussion here, while remaining true to the overall paper scope and purpose (as described above). We have therefore added a new section 3.3 Contributions of FRIDA's methodology for representing climate damages (the prior section 3.3 on Adaptation has been incorporated into 3.2) in the discussion to explicitly discuss this point, which sets out this framework's contribution to IAM impacts literature, and explicitly discusses the wider use of the damage functions from L978:

Lastly, the general nature of the impact functions in FRIDAv2.1 (i.e. being driven by globally averaged climate variables) allows for their usage in other frameworks and IAMs. This is particularly the case for the literature-based functions, which are externally validated and generally linked to widely used concepts or variables (such as population and crop yield). This may also be the case for the more bespoke internally calibrated functions provided here. Several functions were aggregated from gridded data, allowing for their adaptation to regional-scale modelling with only minor modifications to the code (see the Code and Data Availability statement for all damage functions and processing).

We have also referred to this explicitly at the end of Section 1, and this is mentioned in the concluding Section 4. To the more general point, we hope the wider reorganization of sections of the paper will facilitate this greater interest.

While a full blown comparison of how different approaches to representing impacts would be a huge endeavour, it would be useful to provide some comparisons against how others have represented the same or similar impacts and how those representations may lead to different results.

We agree there is utility in comparison here, though we chose not to initially to avoid either somewhat arbitrary choices or far too much detail which as you note is not possible. We have now added two paragraphs to section 3.3 which refer to both bottom-up process-based and econometric top-down damage function literature, noting that FRIDA's dynamics may vary due to the global focus and broader set of impact channels:

The FRIDAv2.1 model offers a novel contribution to integrated assessment modelling by embedding multiple climate impact channels within a fully coupled, process-based framework. This approach departs from the two dominant traditions in climate impact representation: sector-specific and top-down frameworks. Sector-specific models such as the CIRA model (Neumann et al., 2020) simulate impacts within individual domains like health, agriculture, and coastal infrastructure, often using detailed empirical models but without coupling to broader system feedbacks. Conversely, top-down approaches, such as those used in empirical GDP damage studies (e.g. Burke et al., 2015), estimate aggregate economic losses from historical temperature–GDP relationships, but typically lack attribution to specific impact channels.

In contrast, FRIDA represents a wider set of climate impacts within a unified, feedback-rich system. To prioritize the broader human-Earth system-wide interactions over regional or local detail, it operates at a simplified global scale (Schoenberg et al. 2025). While this design choice has limitations, the model's inclusion of a wide set of impact channels enables a more comprehensive exploration of climate damages and their systemic interactions, which are often excluded from IAMs.

Knowing it would be very difficult (let's say nearly impossible) to retrieve validation data at a global scale, are there local or regional situations where data exist that could provide some form of validation of the impacts presented? Or could you provide at least verification of how the impacts perform under customized scenarios to demonstrate that they behave as expected and have boundary conditions? It would certainly add to the credibility of the presented impacts and therefore the likelihood that they would be used or replicated by others.

We absolutely agree that showing the function responses to a future scenario is a useful demonstration of their utility. We opted to not detail any scenario results in this paper to avoid detracting from the documentation purpose, but we recognise there is a balance to be struck here. Therefore we have added a new Figure A2, showing the response of each damage function to a baseline scenario in FRIDA, immediately following the panel plot of the damage functions themselves:

Figure A2: Panel plot of the timeseries response of the impact functions under a baseline scenario in FRIDA which reaches 3.5K temperature change and 730ppm CO2 concentrations by 2150 (see Code and Data Availability). Parameter sets shown as in Fig. A1; note that the crop response combines the temperature and CO2 effects here.

We feel this has aided readers' interpretations of the key results, without compromising the coherence of the overall paper focus. We also feel it best to keep this in the appendix to relate it closer to A1, which itself belongs here as the functions are shown in Fig 1 already.

The global nature of FRIDA renders any regional detailing and validation of the impact functions unfortunately not plausible. In terms of general validation, since we utilised existing literature for many of the impact functions, we can only be as valid in our functions as these upstream studies, which we demonstrate reasonable fits to in the appendix plots (and Code Availability for more interested readers).

Line 52 – 'This collection' – what collection? There are examples like this throughout the manuscript where it seems like it is assumed the reader has knowledge that the authors have. Minimize that assumption and increase the inclusivity of the text. Another easy example here is the text on page 5 where it is written "All six non-climate modules in FRIDA …". It would be better to introduce this to the reader as "There are six non-climate modules in FRIDA…", for example, or some other way – hopefully my point is clear.

Apologies for the lack of clarity both regarding the collection and the overall model here. This paper is submitted as part of this collection in GMD documenting the FRIDA IAM <a href="https://gmd.copernicus.org/articles/collection12.html">https://gmd.copernicus.org/articles/collection12.html</a> - we have now detailed this on L52, which is the only place the collection is mentioned in the main text:

The new fully coupled FRIDAv2.1 Integrated Assessment Model (IAM), presented in Schoenberg et al., (2025) and detailed further within this paper collection (on "The FRIDA model"), seeks to place ...

We have also clarified your example sentence and other instances in the paper to remove reliance on familiarity with FRIDA (e.g. L985 at the start of the Conclusions; *This paper has described the implementation of the 16 climate impact channels across 12 categories in the FRIDAv2.1 IAM (Fig. 1).*).

Line 52 – there is some switching between components and module when describing FRIDA, which was confusing – especially out of the starting gate. Please revise and look to increase consistency throughout.

In general we tried to navigate the connection between the framing of the FRIDA IAM and the real world it seeks to represent by concretely referring to the Modules of FRIDA, but not using this terminology to describe the real world; this is the case on L52. We absolutely appreciate that this was not done clearly here and elsewhere, so we have adjusted this sentence (see above), which we believe will clarify this distinction through the paper, and we have adjusted other instances in the paper (e.g. L863 where we now ensure to refer to modules in FRIDA rather than components).

Line 54 – 'These climate impact interactions...' Again, an unclear reference to impact interactions not explicitly mentioned in the previous two sentences. Overall, watch the backward referencing with 'these', 'this', and 'those' without explicit labels used before and alongside the backward references. It forces the reader back to previous text and slows the reader down.

This is a useful pointer; we have altered this instance to use "Such", which we hope conveys some abstractness here to point the reader towards future detail rather than backwards. We have also made alterations to other sentences with backwards referencing to aid the flow of the manuscript (e.g. L208 "These channels were".).

Line 60 – "Climate impact channels" – generally fine, but not a widely used term that I've seen. Would be useful to describe what a channel is for the reader.

We agree with this; we used this terminology to refer to specific forms of climate impact (following Piontek et al., 2021), and have described this concept now on L60:

Such interactions are key drivers of the system's overall development, and are individually referred to within FRIDA as "climate impact channels". Each channel denotes a single coherent

process-based mechanism through which changes in the climate system influence another part of the coupled system (Piontek et al., 2021).

There is a lot of integration of table and figure references in the text as nouns or focal points/subjects. Using table and figure references in this way degrades the quality of the writing and makes it boring like a textbook. Instead the focus should be on making meaningful and impactful statements and a strong narrative that is supported by tables and figures.

We agree that this will help with readability – we have incorporated this by adjusting the phrasing, including on figure and table references so these are rarely given outside brackets, aiding the paper flow (e.g. L212 "parameter ranges (Table 1) for these channels").

Table 1 – It would seem the table would be more useful, and the research more replicable and transparent, if the actual equations used were included (where possible) in the table.

We originally implemented equations in the table while drafting the manuscript, but we found that the table became too clunky (as the wide column needed to avoid multiple rows lengthened the overall table somewhat). We believed the relatively simple forms used generally would enable readers to follow the narrative, whilst still getting the approximate form, and motivated readers would then be able to refer to the full equations in the supplement if needed. We hope this strikes a balance between clarity and detail in the main text.

Line 175 – "... uncertainty was re—sampled" How was this done? How did your resampling deal with covariance and "capture the impacts of covariance" (line 177). Please provide more details.

Generally when we build uncertainty ensembles in FRIDA, we sample each parameter individually using Sobol sequencing. But because we use literature estimates for these impact channels, we needed to reproduce their uncertainty ranges. We therefore fit skewnorm distributions to the output of the functions and sampled uniformly within those, to generate 11 parameter sets. We've clarified this now within the main text in this paragraph (L214):

For literature-based estimates, where possible, uncertainty was re-sampled using skewnorm distributions to generate 11 parameter sets with the overall impact equally spaced from the 2.5th to 97.5th percentiles (designated as "Sampled Percentiles" in the Uncertainty column of Table 1). This approach was chosen over independent sampling of each parameter to capture the impacts of covariance between parameter values within each impact channel, and to allow for the independent varying of uncertainties within future uncertainty analysis.

Section 2.1.1 – It seems like a heavy emphasis on Yalew et al. (2020). Are there other sources to provide additional support?

Yalew et al. (2020) conducted a systematic literature review, giving us confidence in their overall conclusions. We found their literature compilation as part of our literature dive to find papers we could use in our analysis, with only van Vliet et al (2016) being consistent with our approach. We did not find other suitable papers which were not included in their systematic review.

Several sentences in the document start with "As discussed/detailed in Section #.#.#..." If it was already discussed then it probably doesn't need to be discussed again. Regardless, every time you start a sentence off like this then the reader starts to think or look back in the document, which slows them down. Make the non-redundant statement and then provide support via

citation/reference, which may include another section. Suggest reducing redundancy and limiting those types of references throughout.

This is another very useful suggestion to aid the flow of the paper; we have incorporated this such that no sentence now starts with this type of phrase.

Line 321 – HFC not spelled out in full in the manuscript.

Thanks for flagging this – we have defined this here now.

Line 599 – "... with FRIDA implementing a process-based, bottom-up approach." There is little if any evidence that FRIDA works this way in the manuscript. There is no discussion of heterogeneity or variation, aggregation, interaction or other similar themes typically found in bottom-up based approaches (whether equation-based, agent-based, or some other form). Without some actual description of FRIDA in the paper these cursory comments about FRIDA feel unsupported and distracting. If more description of FRIDA was provided and then in this case the Land Use and Agricultural components were provided in more detail then this section would be substantially more interesting. Again, especially if some corresponding results were provided later.

Apologies for the lack of clarity on this point. We meant to convey that since FRIDA operates in the system dynamics approach, it necessarily follows a bottom-up approach as opposed to top-down, in the framing of Piontek et al. (2021), specifically their framing that "Bottom-up approaches quantify impacts specifically for individual impact channels.... By contrast, in top-down approaches, climate damages are quantified by econometrically estimating aggregate impacts on economic output". We have clarified this point now on this line (L665), adding also that this is a simplified version of this approach. While this applies to the whole model, as detailed in the FRIDA overview Schoenberg et al (2025) paper we cite (now accepted in this GMD collection) we have clarified that we refer to the climate impacts specifically here.

Line 663 – would love to see some more details about the management activities.

Relatedly to the above discussion, we strive here to separate out climate impacts from the internal workings of the impacted systems, though this split is of course not clean, and even less so in the areas of agriculture and adaptation. Documentation of different FRIDA modules is at different stages of completion, with the Land Use and Agriculture documentation not completed currently; however, when this is available interested readers will be able to refer to that for more details on the internal dynamics. In this version of FRIDA, management practices are not endogenously modelled but are implicitly assumed to diversify. Therefore, practices like tillage, mulching, crop rotations or cover crops are not captured. However, recent model development has incorporated mulching as a form of sustainable agriculture with effects on soil carbon and soil moisture – not in v2.1 but for inclusion in v3.0.

We fully appreciate that this split in information between papers through time is less than ideal, especially during the review process, but we can't see a realistic alternative – we ensured the overview paper was available first to try and minimise these issues.

Line 689 – Please don't start a section off with "as discussed in some other section." Substantial writing improvement to engage the reader is needed.

We fully agree with this; in keeping with an above comment this sentence was already altered to aid the paper flow, amongst other changes (e.g. L834 "the use of conceptually consistent

literature choices (Section 2). While FRIDAv2.1 contains many key impact channels, many are excluded, and while some may be incorporated in future development, several will likely not be contained in the FRIDA modelling framework (Section 3.1).".

Line 766 – "... for several important channels this was not possible." – which ones? Give the reader the specifics.

We have referred to Table 1 now here, to point the reader to the relevant impact channels: "...for several important channels this was not possible (those with "N/A" in the Reference column of Table 1)...".

Line 806 – here you talk about the short runtime of FRIDA but you haven't given any quantitative data to support it or any comparisons with other software. Therefore, it's meaningless since your thoughts about length of runtime will most likely differ from the readers.

We have noted some details on the runtime now in this sentence, including noting the model is set up on a HPC for much larger ensembles (L894):

The flexible nature and short runtime of FRIDA (a few seconds to run 1980-2150 on a standard laptop, with large ensembles also set up on a high-performance computer) means that...

We believe this is sufficient to support this point that we can run large statistical ensembles with FRIDA to explore uncertainties in e.g. tipping points.

Piontek, F., Drouet, L., Emmerling, J., Kompas, T., Méjean, A., Otto, C., Rising, J., Soergel, B., Taconet, N., and Tavoni, M.: Integrated perspective on translating biophysical to economic impacts of climate change, https://doi.org/10.1038/s41558-021-01065-y, 2021.

**Reviewer 3**

The study by Wells et al focusses on the representation of climate impact channels in the FRIDA model. It introduces the sectoral damage representations, damage functions, and discusses limitations and opportunities for future development. Fully appreciating the implications of those decisions requires a comprehensive understanding of the full FRIDA model. I studied also Schoenberg et al (in discussion) to get a better feel for how these impacts may represent model dynamics, but some parts remain rather elusive to me.

I want to start by applauding the authors for a massive effort. I enjoyed reading the manuscript and find the approaches well explained and state of the art.

However, I have a few remarks relating to the overall approach as well as to some sector specific outcomes.

Many thanks for the kind words and taking the time for this detailed review of our manuscript. We have made alterations on the basis of these which we believe substantially improve the paper; we have detailed these below.

The choice of the impact channels. To some extent the impact channels you model are the 'easy' ones, many of which are represented at least partly in complex IAMs (i.e. energy system impacts, cooling needs, crop yields, etc). Not all, of course, but many. Which raises a bit the question of where the major innovation lies in the FRIDA coupling. And then there are some surprising omissions, such as forest fires, that I would have expected to feature more prominently. Also, the representation of damage functions is straight forward and mostly linear. Which is how it is in most models, but when building a model from scratch today, I would have given some thought on how to detect potential non-linear thresholds etc.

We agree that the specific selection of impact channels is subject to some potentially arbitrary conditions. However, we tried to be explicit and reflective about our methodology for selecting climate impacts, which was constrained by the existing literature and the inherent properties of the modelling framework, as well as the model scope, in Section 3.2. We agree that we should have been more explicit on the novelty of the approach in FRIDA; we have added a few new sentences around L147 to present this, noting that in FRIDA we seek to treat all inter-system feedbacks and processes at the same level of detail:

FRIDA's novelty lies in the overarching philosophy of treating climate impact channels as just a subset of the overall inter-system feedbacks across all components of the human-Earth system, conceptualised in FRIDA as its modules. As such, process-based feedbacks are sought to be represented at similar levels of detail across the model. In addition, recognising the conceptual issue in frameworks such as the SSPs in neglecting impacts, FRIDA's approach aims to represent as broad a set of climate impacts as reasonably possible, a goal which will continue to advance under future model development.

Regarding forest fires, we agree this was a borderline case which we could have explored in this version. We ultimately chose not to due to the complexities of the wildfire nexus, in terms of human management, ignition, regional variation, and climate impacts, the latter especially since we do not model non-sulfur aerosols currently. A representation of fires is currently being implemented to go into v3.0 of FRIDA, connecting burned area to the level of climate change, but complexities remain on the full effects (we note this briefly in Section 3.1 around L775).

Regarding the simplicity of damage functions: we agree that more complex forms, including tipping points, could be explored, with tipping points indeed an area of current investigation in the model as discussed in the manuscript, but these were not explored in v2.1 due primarily to the uncertainties involved in the literature, especially in the downstream impacts of tipping. We however would like to note that many of the damage functions are nonlinear, with many represented quadratically, allowing for changing sensitivity at different warming levels; we have now noted this explicitly on L891. Also, the feedback-based approach in FRIDA allows for nonlinearities in the overall system due to component interactions (see discussion around L901). As noted in the new paragraph above, FRIDA's novelty also rests on its inclusion of this wide set of impact channels in one model.

I would have also liked to see a sensitive testing to some of the assumptions to understand which impact channels are the most impactful ones in terms of full century trajectories (I'd suspect the least well constrained ones on government spending and indirect effects, but can't proof this

We highlight the new figure A2 introduced in response to reviewer 2's comments, showing the response of each climate impact channel under a baseline emissions scenario:

Figure A2: Panel plot of the timeseries response of the impact functions under a baseline scenario in FRIDA which reaches 3.5K temperature change and 730ppm CO2 concentrations by 2150 (see Code and Data Availability). Parameter sets shown as in Fig. A1; note that the crop response combines the temperature and CO2 effects here.

We agree in principle that it would be useful to explore future scenarios in this paper; indeed we did incorporate this in our initial drafts using a prior version of the model, in which we did find an important role of the economic effects, as you suggest. But we decided against incorporating this here in more detail than Figure A2, for several reasons: firstly due to the very large size of the

paper, which we felt could become unwieldy and distract from the damage functions themselves; secondly to keep to the scope of description of the impact channels and the process of their construction; and thirdly since we plan to dig into this complex issue in a standalone study, allowing us to fully explore the detail in this area, which we would not be able to do here. We appreciate there is a balance to be struck here and hope that our justifications here are suitable to keep the focus as it currently stands.

This point relates to the FRIDA development more generally, but is also relevant here. There are models of this class out there, such as the FELIX model developed for more than a decade now. They attempt to do very similar things, and have very similar limitations. Yet, there are not even cited nor is some comparison performed. This is a bit unfortunate, because quite some learnings could have been taken from this class of models also in relation to the specific FRIDA innovations (i.e. in relation to the probabilistic modelling).

We absolutely agree on the need to situate FRIDA more closely within the existing landscape, and we thank you for highlighting this oversight here. We do have connections to the FeliX team, and we refer to this model and others in the final version of the Schoenberg et al overview paper. We now refer to the Schoenberg et al paper specifically on this point, which we feel is the appropriate approach given the broader discussions within that paper, and the focus of this one (L55):

FRIDA is designed as a system dynamics-based coupled human and natural systems (CHANS) model building on prior system dynamics approaches to human-Earth modelling (Schoenberg et al, 2025).

Regarding other IAM approaches to climate damages, we have added new paragraphs on this to the discussion section in this paper (L958):

Sector-specific models such as the CIRA model (Neumann et al., 2020) simulate impacts within individual domains like health, agriculture, and coastal infrastructure, often using detailed empirical models but without coupling to broader system feedbacks. Conversely, top-down approaches, such as those used in empirical GDP damage studies (e.g. Burke et al., 2015), estimate aggregate economic losses from historical temperature—GDP relationships, but typically lack attribution to specific impact channels.

In contrast, FRIDA represents a wider set of climate impacts within a unified, feedback-rich system. To prioritize the broader human-Earth system-wide interactions over regional or local detail, it operates at a simplified global scale (Schoenberg et al.. 2025). While this design choice has limitations, the model's inclusion of a wide set of impact channels enables a more comprehensive exploration of climate damages and their systemic interactions, which are often excluded from IAMs.

Some of the impact channels remain a bit opaque to me. The first one that comes to mind is the demographic one. From what I understand it's only mortality that's modelled, but for a model of that class, other channels, in particular on education may proof much more relevant. I understand from Schoenberg that the female education – pop-growth channel is explicitly modelled. Which means it could be really quite relevant to explore climate impacts on this for 21st century population growth. It seems FRIDA pop growth is close to SSP3 – but with limited variance compared to other variables.

We apologise if the manuscript is unclear on the detail of some impact channels. We have tightened up the manuscript on suggestions from Reviewer 2 (especially sections 2.1.1, 2.1.2, 2.2.1), and expect this aids here. We agree on the importance of demographic modelling, but we are unfamiliar with previous studies connecting climate to educational outcomes; as you note we have a (simplified) representation of broader education dynamics within that module, but we only document the climate impacts here, which as you note are only targeted on mortality currently.

Similarly, the economic damages modelling is not very clear to me. The authors want to endogenize economic damages, but then assume so fairly simplistic scaling functions of government spending etc. From Schoenberg et al. Fig 10 I take it that economic growth in FRIDA is at the low end of the SSP scenarios. Is this because of climate impacts, or just the economic module of FRIDA? And how do the endogenized impacts compare to other damage functions? I think it would be very useful to understand how FRIDA's economic performance would look like with and without climate damages and how it compares to macro-economic damage functions in the literature.

The impact of the economic damage functions (indirect effects and government spending) are designed to be real-world realistic to the KPIs they impact and are based on the literature we cite. As suggested in Schoenberg et al., 2025, it is these climate-driven changes in government budget allocations and loan defaults that go on to create the economic consequences in Figure 10 of that paper. In this paper our purpose is to document the specific structure of the impacts, and the literature upon which they are based. In a forthcoming paper on the economic system of FRIDA we will explain in detail how this climate damage mechanism is incorporated into FRIDA's economic system (beyond what has been done here, and in Schoenberg et. al, 2025), and in a second forth-coming paper we will present a study of the causal mechanism that activates the feedback processes within that economic structure that creates the outcomes seen in Figure 10. That second forthcoming paper compares the impact of these damage function representations to a typical DICE style aggregate economic damage function. In our view these deeper explanations cannot be included within this paper due to length and scope.

Treatment of adaptation and adaptation costing: I find it reasonable that if the authors had to pick one sector for adaptation it should be sea level rise. Yet, for near-term economic and societal damages, arguably heat related mortality and morbidity all the way to labour productivity are probably more impactful. And somewhat of a miss that adaptation is not coupled to macro-indicators resembling adaptive capacity. The authors cite the Andrijevic et al work, and from what I take their model would fully allow them to endogenize the adaptive capacity modelling also. I understand it's an avenue for future work, but could be useful – coupled with the government spending module, for example.

We absolutely agree that it will be a crucial part of future FRIDA development to incorporate channel-specific adaptation strategies, including potentially via a framing of adaptive capacity. However, in the process of the construction of FRIDAv2.1, we opted to first build up a set of climate impact channels, with only a limited focus on adaptation where deemed most crucial on SLR. It is absolutely the case however that this will cause greater issues for near-term simulations than longer-terms ones due to the dynamics of SLR; we have now noted this fact briefly in Section 3.2 L933: "...few active adaptation mechanisms are represented in FRIDAv2.1, with gaps particularly in channels with strong expected near-term impacts." We hope this is sufficient, and we take this comment as further motivation for a renewed focus on adaptation in future model development.

**Minor comments:**

Table 1: Entry for HDD. C&P mistake in the description. I assume it's "heating energy" that's required here.

Many thanks for spotting this! We have fixed this now.

L540: I'm not sure this comprehensive enough in terms of 'indirect economic effects'. Expectations is an important part, but other dimensions link to dampened economic activity as ripple on effects of destroyed assets or infrastructure e.g. linked to extreme weather events. This may be subsumed here, but it also wouldn't hurt to call it out a bit more concretely.

We agree that these cascading ripple effects are a key component of the economic effect which we endeavour to account for here. We have now clarified this paragraph (around L604) to make this point:

The strengths of these indirect economic effects are calibrated as part of the full-model calibration, using past data. This implies that these indirect economic effects are at work for the entire range of the model simulation. Indirect economic effects—are mediated on one hand by expectations, whereby as evidence for future climate risks increases, reducing uncertainty, their weight in economic calculation similarly increases; and on the other hand by cascading effects, with destroyed assets in one region or sector triggering bankruptcies in entities that are dependent upon those destroyed assets. Consequently, while logically secondary to other climate damages, the compounding, forward-looking nature of these effects implies that their economic consequences are significant.

L700: I am a bit confused that impacts on forest fires, are not more prominently explored, and that the carbon cycle response is probably largely the representation in FaIR. In such a coupled whole-system model, they strike me as some of the key feedbacks one would like to capture.

Please see above comments regarding forest fires, which we recognise the importance of but excluded from FRIDAv2.1 due to their complexity and uncertainty. We would like to clarify however that the carbon cycle response is not the simple box-model in FaIR; see the discussion around L906 "These occur in the carbon cycle…" – we have made a process-based carbon cycle representation in FRIDA, which includes several internal feedbacks (which we don't categorize as climate impacts but we describe briefly here and in the Climate Module documentation, now submitted to this collection). This process-based carbon cycle allows the possibility of coupling fires in as part of future development.

**PS:**

Apologies to the authors for taking so long to complete my review. It's a very busy time of the year and this a very comprehensive piece of work. I commend their efforts and want to express my regrets of slowing down the review process.

Absolutely no worries at all! We really appreciate the time taken to review this article.